# Spatially Aware Linear Transformer (SAL-T) for Particle Jet Tagging

## Abstract

Transformers are very effective in capturing both global and local correlations within high-energy particle collisions, but they present deployment challenges in high-data-throughput environments, such as the CERN LHC. The quadratic complexity of transformer models demands substantial resources and increases latency during inference. In order to address these issues, we introduce the Spatially Aware Linear Transformer (SAL-T), a physics-inspired enhancement of the linformer architecture that maintains linear attention. Our method incorporates spatially aware partitioning of particles based on kinematic features, thereby computing attention between regions of physical significance. Additionally, we employ convolutional layers to capture local correlations, informed by insights from jet physics. In addition to outperforming the standard linformer in jet classification tasks, SAL-T also achieves classification results comparable to full-attention transformers, while using considerably fewer resources with lower latency during inference. Experiments on a generic point cloud classification dataset (Model-Net10) further confirm this trend. Our code is available here.

## 1 Introduction

Attention-based transformers (1) are ubiquitous in machine learning applications from natural language processing to computer vision. Given their prominence and state-of-the-art (SOTA) performance, they have also been applied in the physical sciences, including high energy physics, where they have usurped the previous SOTA architectures like convolutional, recurrent, and graph neural networks. Cutting-edge scientific experiments like those at the CERN Large Hadron Collider (LHC) (2) address fundamental questions in particle physics and rely on transformers to maximize insights from the data collected. The data produced by these experiments contain information about the particles produced in proton-proton collisions and can be represented as point clouds. Transformers can discern the intricate correlations in particle showers, known as *jets*, caused by the decay of heavy particles, allowing physicists to identify these particles (a task known as *jet tagging*), search for new particles or interactions, and measure the properties of known particles and interactions (3; 4).

Despite this success, one major drawback of transformers is their computational complexity. The attention mechanism requires $\mathcal{O}(n^2)$ operations for $n$ input tokens. Typically, the size of each layer and the number of heads are also large, resulting in a large number of floating-point operations (FLOPs). One of the critical applications of neural network-based jet tagging is in the real-time online collision event filter known as the trigger (5; 6). The LHC collides protons 40 million times per second, forming point clouds of detector measurements. Due to storage limitations, only about 1 in 40 000 events are stored for further analysis using real-time algorithms suitable for high data throughput (7; 8). Due to their computational complexity, transformers cannot be readily employed in this online data filtering process.

In recent years, several works have explored reducing the computational complexity of transformers. This includes replacing the attention mechanism with low-rank (9) approximations. The low-rank approximations decrease computation complexity, but this could come at the cost of performance. However, designing a linear attention method utilizing the underlying structure could improve performance while reducing the computation cost.

Looking ahead, the High-Luminosity LHC (HL-LHC), expected to begin collisions in 2031, will provide an unprecedented environment to develop and deploy advanced machine learning algorithms at trigger level. With increased luminosity and correspondingly higher event rates, more sophisticated real-time filtering will be essential. The availability of Particle Flow (PF) (10)candidates, including their azimuthal angle and pseudorapidity relative to the jet axis, as early-stage features in the firmware pipeline, offers new opportunities for designing efficient and physics-aware algorithms. These experimental conditions motivate research into architectures that can balance accuracy with stringent computational and memory constraints, paving the way for next-generation approaches to jet tagging at the HL-LHC.

**Related Work**   State-of-the-art deep learning efficient models for sequence processing—such as the longformer (11), performer (12), linformer, and reformer (13)—are primarily optimized to handle very long input lengths (tens of thousands of tokens) by sparsifying or approximating full self-attention. While these methods achieve sub-quadratic complexity in $n$, they introduce specialized kernels and overhead that often outweigh their benefits when applied to moderate-length inputs ($n \approx 100$ tokens) that are common in particle physics scientific workflows.

Particle transformer (3) is a SOTA jet tagging model, which introduces a modified attention mechanism, called particle multi-head attention (P-MHA), that considers pairwise features as an attention bias. A similar architecture has been developed using only pairwise features for attention, which reduces the computational resources, but maintains the $\mathcal{O}(n^2)$ scaling (14). A locality-sensitive hashing-based efficient point transformer (HEPT) has also been developed and applied to charged particle tracking and pileup mitigation in high energy physics, with improvements demonstrated for token lengths between $6\,000$ and $60\,000$ (15). Transformers based on induced self-attention (16) have also been explored for generative modeling of jets (17; 18). Despite their SOTA performance on particle physics tasks, the latency and resource restrictions in the trigger prevent deployment of these models. In this paper, we address this gap by introducing a spatially-aware attention model that learns geometric relations and spatial features, while lowering compute cost when compared to transformer.

## 2   SPATIALLY AWARE LINEAR TRANSFORMER

We propose the spatially aware linear transformer (SAL-T) as an efficient and physics informed alternative to traditional transformers for jet tagging. In the context of LHC physics, jets are collimated sprays of particles resulting from the hadronization of quarks or gluons. To characterize these jets, we use a Cartesian coordinate system with the $z$ axis oriented along the beam axis, the $x$ axis on the horizontal plane, and the $y$ axis oriented upward. The azimuthal angle $\phi$ is computed with respect to the $x$ axis. The polar angle $\theta$ is used to compute the pseudorapidity $\eta = -\log(\tan(\theta/2))$. The transverse momentum ($p_{\mathrm{T}}$) is the projection of the particle momentum on the $(x, y)$ plane. For each particle in the jet, we compute the relative angular coordinates $\Delta\eta$ and $\Delta\phi$ relative to the jet axis. Jets exhibit spatial structure in the detector, characterized by the distribution of particles in $\Delta\eta$ and $\Delta\phi$. In addition, the particles with the greatest $p_{\mathrm{T}}$ are generally the most directly related to the original decaying particle.

Linformer encodes no spatial information making it indifferent to the substructure of jets, limiting its effectiveness where spatial correlations carry critical information. To overcome this, SAL-T introduces spatial awareness through three modifications to linformer: In our approach, we (1) sort the input particles by spatial proximity in the $(\Delta\eta, \Delta\phi)$ plane, weighted by transverse momentum $p_{\mathrm{T}}$, to ensure that physically relevant, nearby particles are close to one another in the sequence, (2) partition the key and value projections into $p$ groups so that each projection head attends only to its own subset of particles, and (3) enhance the attention map by applying a small depthwise 2D convolution over each head's raw attention scores to incorporate local neighbor interactions without reintroducing quadratic complexity. These yield our core attention module: linear partitioned particle multi-head attention (LPP-MHA).

## 2.1 LINEAR PARTITIONED PARTICLE MULTI-HEAD ATTENTION

Let $X \in \mathbb{R}^{n \times d}$ represent the input sequence of $n$ particles, each with $d$-dimensional features. We compute the standard query, key, and value matrices:

$$Q = XW_Q \in \mathbb{R}^{n \times d_k}, \quad K = XW_K \in \mathbb{R}^{n \times d_k}, \quad V = XW_V \in \mathbb{R}^{n \times d_k}, \tag{1}$$

where $W_Q, W_K, W_V \in \mathbb{R}^{d \times d_k}$ are learnable projections.

**Locality Aware Sorting and Partitioning** To ensure each projection focuses on a physically coherent subset of particles, we sort the sequence $X$ by in descending order by the theoretically motivated metric, $k_\mathrm{T} = p_\mathrm{T} \Delta R$, where $\Delta R = \sqrt{(\Delta \eta)^2 + (\Delta \phi)^2}$ is the pseudoangular distance to the jet axis (19). The $k_\mathrm{T}$ metric is larger for particles that have larger transverse momentum or are more closely aligned with the jet axis. This metric is used in iterative jet clustering algorithms at particle colliders to produce jets with the theoretically desirable properties of *infrared safety*—meaning the jet is insensitive to the addition of arbitrarily low-energy particles—and *collinear safety*—meaning the jet is insensitive to the splitting of a particle into two or more particles that are moving in nearly the same direction (19; 20; 21; 22). As a result, partitions derived from $k_\mathrm{T}$-sorted sequences are more likely to group together energetic particles that are physically nearby, enhancing the effectiveness of spatially aware projection, as illustrated in Fig. 1. The default sorting in LHC physics is descending order by $p_\mathrm{T}$, which does not guarantee adjacent particles are spatially local. This limitation is illustrated in Fig. 1, where $p_\mathrm{T}$-based sorting can lead to spatially distant particles appearing adjacent in the sequence. SAL-T is the first algorithm designed for trigger usage that takes advantage of particle sorting. Many production trigger algorithms, such as AXOL1TL and CICADA (7), are not permutationally equivariant, yet they do not exploit the sorting information available in the detector. We define our sorting as:

$$X = \mathrm{Sort}\big(X_{\mathrm{unsorted}}, \ \mathrm{key} = \|\mathbf{k}_\mathrm{T}\|\big) \tag{2}$$

The key and value vectors are then partitioned into $p$ partitions:

$$K_P^{(i)} = \begin{bmatrix} K_{(i-1)\frac{n}{p}} \\ K_{(i-1)\frac{n}{p}+1} \\ \vdots \\ K_{i\frac{n}{p}-1} \end{bmatrix}, \quad V_P^{(i)} = \begin{bmatrix} V_{(i-1)\frac{n}{p}} \\ V_{(i-1)\frac{n}{p}+1} \\ \vdots \\ V_{i\frac{n}{p}-1} \end{bmatrix} \tag{3}$$

where $K_P^{(i)}, V_P^{(i)} \in \mathbb{R}^{n//p \times d_k}$, $i = 1, \dots, p$ denotes the partition number, and the key and value vectors are sliced along the sequence dimension. We define learnable projections $P_E^{(i)}, P_F^{(i)} \in \mathbb{R}^{1 \times n//p}$ which individually act on each $X_P^{(i)}$,

$$K^P = \mathrm{concat}_{i=1}^p \big(P_E^{(i)}, K_P^{(i)}\big), \quad V^P = \mathrm{concat}_{i=1}^p \big(P_F^{(i)}, V_P^{(i)}\big),$$
$$K^P, V^P \in \mathbb{R}^{p \times d_k}, \quad i = 1, \dots, p. \tag{4}$$

Each projection row in $P_E$ or $P_F$ only attends to one such partition, ensuring locality. Each projected vector is then restricted to aggregate embeddings only from its corresponding partition, as illustrated in Fig. 1. Each partition will only be superpositions of $n/p$ particles, with any leftover padding where the number of particles does not reach the maximum number of allowed particles in the jet will be located in the last partitions. This spatially structured design allows SAL-T to capture local jet substructure while maintaining linear complexity in sequence length, enabling efficient and physically-informed modeling.

The projections $P_E$ and $P_F$ are designed such that each row of the output aggregates only over a localized partition of the input. In our design, for the projected vectors to preserve local substructure information relevant to jet tagging, each partition must correspond to a localized group of particles in the $(\Delta \eta, \Delta \phi)$ plane. Since $P_E$ and $P_F$ project the input sequence from $n$ dimensions to $p$ dimensions, the computational complexity reduces from $\mathcal{O}(n^2)$ to $\mathcal{O}(np)$, which is near linear when $p \ll n$. By construction, this would mean that for jets with much fewer particles than the sequence length of $n$, only the first partitions will be used, while the other partitions will be populated with zeros. On the other hand, when the number of particles in the sequence is near $n$, all partitions will be used. This allows the model to conserve resources for smaller jets that do not have as much relevant substructure, while focusing in on relevant substructure for larger jets.

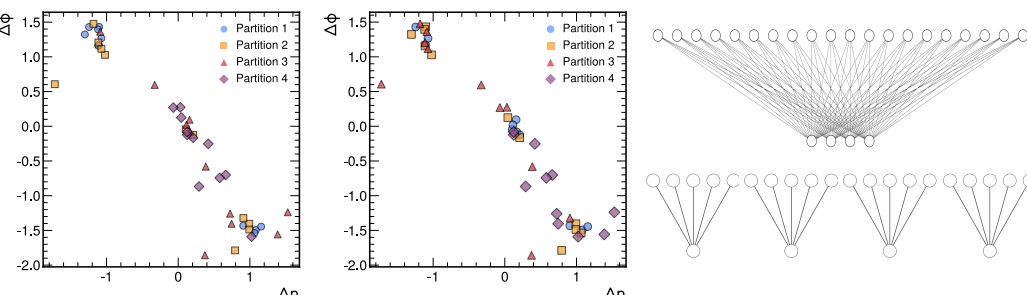

Figure 1: (*Left*) Jet constituents partitioned and sorted by $k_T$ in the $(\Delta\eta, \Delta\phi)$ plane in SAL-T, showing how constituents are binned spatially before projection. (*Center*) Jet constituents partitioned by transverse momentum in the $(\Delta\eta, \Delta\phi)$ plane. (*Right*) Visualization of the projection partitioning strategy used in SAL-T, Jet constituents are partitioned into spatial bins before projection, preserving local structure.

**Convolution**   Convolutional layers can be applied directly on top of the attention matrix to allow for more context-rich attention (23). In particle jet datasets, this would allow the attention weights of each particle to impact the attention weights of nearby particles. When sorted by $k_T$, a convolutional filter would look at a group of particles close together in physical space, allowing the attention to learn more spatial context. After the embeddings are split into $H$ attention heads with dimensionality $d_h = d_k/H$, we perform a depthwise 2D convolution on the attention matrix logits within each head, where each filter is applied along the sequence axis, spans the entire projection dimension, and uses "same" padding to preserve the output dimension. The filter responses are averaged and passed through a softmax function to yield an attention matrix that incorporates local particle information. Because these convolutions work on the low-dimensional per-head channels with fixed kernels, they add only a linear cost in sequence length. This module allows attention to leverage immediate neighbors in the $k_T$-sorted sequence, capturing spatial patterns without reintroducing quadratic complexity. Concretely, LPP-MHA is computed as

$$\text{LPP-MHA}(X) = \text{concat}_{h=1}^{H} \left[ \text{softmax} \left( \text{conv2d} \left( Q_h K_h^P / \sqrt{d_h} \right) \right) V_h^P \right], \quad (5)$$

and its computational complexity is $\mathcal{O}(n\,p\,f\,c)$, where $n$ is the sequence length, $p$ is the projection dimension, $f$ is the number of filters, and $c$ is the kernel width.

## 3 EXPERIMENTS

To assess the suitability of our spatially aware linear transformer (SAL-T) for deployment in latency- and resource-constrained environments such as the trigger, we constrain each model to at most two multi-head attention layers and limit the total number of trainable parameters to a few thousand. These restrictions ensure compatibility with the stringent timing and memory requirements of trigger systems. We evaluate each model's hardware efficiency—measured via resource utilization and inference latency-alongside classification performance, area under the receiver operating characteristic curve (AUC) and background rejection, to determine viability for real-time triggering applications. Background rejection refers to the inverse background efficiency at a fixed signal efficiency of 80%, denoted as $1/\varepsilon_B(\varepsilon_S = 0.8)$. This metric quantifies a model's ability to suppress background events while maintaining moderate signal retention, and serves as a practical measure of its discriminative power in classification tasks.

### DATASETS

The publicly available `hls4ml` dataset (24; 25) contains 504,000 jets in the train set, 126,000 jets in the validation set, and 240,000 jets in the test set. There are five classes of jets, each class originating from the decay of the following particles: light quark (q), gluon (g), W boson, Z boson, and top quark (t), each represented by the four-momentum vectors of up to 150 constituent particles. We process the dataset to best replicate the conditions in the trigger. We use the same setup as in Ref. (26). The

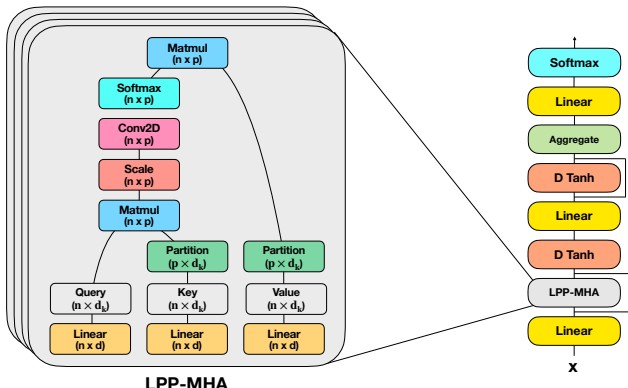

Figure 2: (*Left*) Architecture of the linear partitioned particle multi-head attention (LPP-MHA) module used in SAL-T. The input query, key, and value sequences of dimension $n \times m$ are linearly projected to dimension $n \times d$, then spatially partitioned into $p$ groups of size $p \times d$. Attention weights are computed via scaled dot-product attention within each partition, followed by a depthwise convolution over the attention map to promote local context mixing. The resulting attention matrix is used to aggregate value representations, forming the basis for the output of the attention layer. This design maintains computational efficiency while maintaining locality-aware expressivity. (*Right*) One layer SAL-T model).

initial-state particles are generated to have a $p_\text{T}$ of at least 1 TeV, while the final state energies are smeared to replicate conditions in the CMS detector. We then take the $n$ highest particles with a $p_\text{T}$ of greater than 1 GeV. When there are less than $n$ constituents, we pad the jet with zeros until it is of length $n$. We use transverse momentum ($p_\text{T}$), pseudorapidity ($\Delta\eta$), and azimuthal angle ($\Delta\phi$) relative to the jet axis as the input features. The $p_\text{T}$ is normalized to its corresponding quartile range of 5 and 95%.

Beyond the `hls4ml` dataset, to demonstrate the robustness of SAL-T, we also conduct experiments on two binary classification datasets: the Top Tagging dataset (27; 28) and the Quark Gluon dataset (29; 30). The Top Tagging dataset contains 1.2 million jets in the training set, 0.4 million jets in the validation set, and 0.4 million jets in the testing set. There are two classes of jets: signal jets originate from the decay of top quarks, and background jets originate from light quarks or gluons in dijet events. The Quark Gluon dataset contains 1.8 million jets for training and 0.2 million jets for testing. Out of the 1.8 million training jets, we randomly sampled 20% for validation. In the Quark Gluon dataset, there are two classes of jets, originating from quarks and gluons, respectively.

In addition, we evaluate SAL-T on a generic point cloud classification benchmark: the ModelNet10 dataset (31). ModelNet10 contains 4,899 training and 908 testing 3D objects from 10 categories of man-made shapes (e.g., chair, sofa, table). For each object, we use the standard protocol of sampling 1,024 points from the mesh surface to form the point cloud input. The task is a 10-way classification over object categories.

MODEL ARCHITECTURES

All models embed the input jet constituents into a 16-dimensional feature space and use $H = 4$ attention heads, followed by a lightweight max-aggregation layer and a final dense classification head. We use dynamic tanh instead of layer norm for normalization across all models due to faster inference with similar performance (32). All models use the same architecture as in Fig. 2.

- **SAL-T:** A single LPP-based multi-head attention layer with embedding dimension $d = 16$ and $H = 4$ heads; keys and values are projected via spatially aware partitioned matrices $P_E$ and $P_F$ (rank 4), and before softmax each head's raw attention logits are enhanced by three 2D convolutional filters (heights 1, 3, and 5; width matching the number of projections) with "same" padding to preserve dimensions.

Table 1: Performance comparison of different models evaluated on jets with up to 150 constituent particles. We report accuracy, area under the ROC curve (AUC), and average rejection (1/FPR at a fixed signal efficiency) as the primary classification metrics. Results are shown for various models and sorting strategies applied to the input sequence, including transverse momentum ($p_T$) sorting and $k_T$-based sorting. Each value is reported as the mean $\pm$ standard deviation across multiple trials. The transformer is permutation invariant so only the ($p_T$) sorting performance is shown.

| Model | Accuracy [%] | ROC AUC | 1/FPR@0.8TPR |
|---|---|---|---|
| SAL-T ($p_T$ sorted) | $78.82 \pm 0.01$ | $0.950 \pm 0.0028$ | $23.45 \pm 2.12$ |
| Linformer ($p_T$ sorted) | $79.90 \pm 0.00$ | $0.9545 \pm 0.0004$ | $28.06 \pm 0.58$ |
| Linformer ($k_T$ sorted) | $81.00 \pm 0.08$ | $0.9585 \pm 0.0003$ | $38.41 \pm 0.54$ |
| SAL-T ($k_T$ sorted) | $\mathbf{81.18 \pm 0.03}$ | $\mathbf{0.9593 \pm 0.0002}$ | $\mathbf{40.78 \pm 0.57}$ |
| Transformer ($p_T$ sorted) | $81.27 \pm 0.08$ | $0.9589 \pm 0.0004$ | $42.02 \pm 0.71$ |
| Deep set | $79.65 \pm 0.23$ | $0.8336 \pm 0.0376$ | $7.41 \pm 0.99$ |
| Interaction network | $80.05 \pm 0.46$ | $0.9128 \pm 0.0139$ | $22.01 \pm 3.69$ |
| MLP | $53.09 \pm 1.32$ | $0.8054 \pm 0.0120$ | $3.66 \pm 0.12$ |

- **Linformer:** Identical to SAL-T in embedding dimension ($d = 16$), head count ($H = 4$), and projection dimension ($k = 4$), but using low rank key value projections (9) without convolutional filters.
- **Transformer:** One standard multi head self attention layer with embedding dimension $d = 16$, $H = 4$, key query value dimension $d_k = 4$.

In all cases, the per token outputs of the attention or LPP MHA layer are aggregated via max aggregation over the sequence and then passed to a final dense layer mapping to the five jet class logits.

**Training** Each model is trained for 1400 epochs using a phased batch size schedule: 200 epochs each with batch sizes of 128, 256, 512, 1024, and 2048, followed by 400 epochs with a batch size of 4096. Optimization is performed using Adam with a learning rate of 0.001. Early stopping is applied if the validation loss (categorical cross-entropy for multi-class tasks or binary cross-entropy for binary classification) fails to improve for 40 consecutive epochs. All experiments are conducted on the National Research Platform (NRP) Nautilus cluster (33; 34) using NVIDIA GTX 1080-Ti or GTX 3090 GPUs.

## 4 RESULTS

To validate the effectiveness of SAL-T, we evaluate five model configurations across three performance metrics on the hls4ml dataset, as shown in Table 1. SAL-T applied to $k_T$-sorted jets outperforms both Linformer baselines on all three metrics and achieves performance comparable to the full quadratic attention Transformer, accounting for statistical fluctuations (values are reported as mean $\pm$ standard deviation across multiple trials). Notably, $k_T$ sorting consistently improves performance over standard $p_T$ sorting for both SAL-T and linformer, demonstrating that spatial ordering increases the amount of contextual information between each particle.

In addition, SAL-T significantly outperforms earlier trigger-oriented models (26) including a multilayer perceptron (MLP), deep sets (35; 29), and interaction network (36; 37), demonstrating its suitability for low-latency jet tagging. For jets with larger numbers of constituents, SAL-T maintains parity with the transformer while outperforming linformer, indicating a greater capacity to capture complex jet substructures.

To assess efficiency, we report hardware benchmarks in Table 2. SAL-T matches the inference efficiency of Linformer-based models and offers substantial improvements over the full transformer in both memory usage and inference latency. Taken together, these results show that SAL-T consistently surpasses linformer in classification performance while matching its hardware efficiency, and achieves performance on par with the full transformer at significantly lower computational cost.

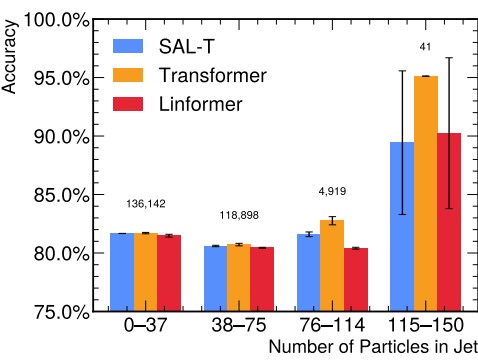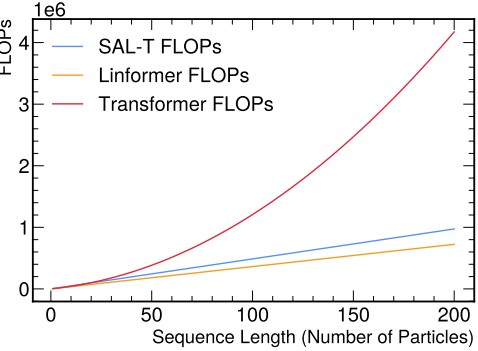

Figure 3: (*Left*) Jet classification accuracy of SAL-T, Linformer, and standard Transformer across bins of increasing number of particles per jet. SAL-T consistently matches or exceeds the accuracy of linformer and remains competitive with full transformers. Performance variance in the highest bin (115–150 particles) is attributable to its small sample size (only 41 jets). (*Right*) Floating-point operation (FLOP) counts as a function of sequence length for the three models. While transformer FLOPs grow quadratically with input length, SAL-T maintains nearly linear scaling, closely tracking linformer while offering improved performance in high-multiplicity jets.

Table 2: Inference benchmarks of different models for the 150-particle setting. Metrics reported include the number of parameters, floating point operations (FLOPs), peak GPU memory usage, and average inference time (with standard deviation) per jet.

| Model | # Params | FLOPs | Peak GPU Mem [MB] | Inference Time [μs] |
|---|---|---|---|---|
| SAL-T | 3,264 | 739,918 | 303.4 | $27.69 \pm 0.32$ |
| SAL-T (no partition) | 6,848 | 739,918 | 296.4 | $27.06 \pm 0.17$ |
| SAL-T (no convolution) | 3,225 | 552,718 | 226.2 | $22.83 \pm 0.84$ |
| Linformer | 6,809 | 552,718 | 245.8 | $22.38 \pm 0.33$ |
| Transformer | 2,009 | 2,479,918 | 4,357.1 | $30.86 \pm 0.14$ |
| Deep set | 3,461 | 664,805 | 220.3 | $17.47 \pm 0.25$ |
| Interaction network | 10,064,702 | 61,529,931 | 4,283.9 | $307.38 \pm 0.44$ |
| MLP | 53,837 | 267,081 | 92.9 | $16.42 \pm 0.25$ |

Table 3: Model performance comparison on the Top Tagging dataset. Best metrics between SAL-T and linformer are highlighted in bold. In the case where two metrics are within statistical uncertainty, neither is highlighted. All models are sorted by $k_{\mathrm{T}}$.

| Layers | Model | # Params | FLOPs | Accuracy [%] | ROC AUC | 1/FPR@0.8TPR |
|---|---|---|---|---|---|---|
| 1 | SAL-T | 3,580 | 986,161 | $92.52 \pm 0.11$ | $0.9780 \pm 0.0006$ | $31.84 \pm 1.10$ |
| | Linformer | 8,341 | 736,561 | $92.40 \pm 0.07$ | $0.9774 \pm 0.0005$ | $30.89 \pm 1.13$ |
| | Transformer | 1,941 | 4,186,161 | $92.91 \pm 0.06$ | $0.9802 \pm 0.0003$ | $36.54 \pm 0.77$ |
| 2 | SAL-T | 6,939 | 2,794,161 | $\mathbf{92.79 \pm 0.03}$ | $\mathbf{0.9796 \pm 0.0001}$ | $\mathbf{35.15 \pm 0.17}$ |
| | Linformer | 16,329 | 1,450,161 | $92.61 \pm 0.02$ | $0.9786 \pm 0.0001$ | $33.11 \pm 0.22$ |
| | Transformer | 3,529 | 8,349,361 | $93.11 \pm 0.03$ | $0.9813 \pm 0.0001$ | $40.11 \pm 0.23$ |

Table 4: Model performance comparison on the Quark Gluon dataset. Best metrics between SAL-T and Linformer are highlighted in bold. In the case where two metrics are within statistical uncertainty, neither is highlighted. All models are sorted by $k_\mathrm{T}$.

| Layers | Model | # Params | FLOPs | Accuracy [%] | ROC AUC | 1/FPR@0.8TPR |
|---|---|---|---|---|---|---|
| 1 | SAL-T | 3,196 | 739,761 | $81.34 \pm 0.05$ | $\mathbf{0.8888 \pm 0.0005}$ | $5.76 \pm 0.05$ |
| | Linformer | 6,741 | 552,561 | $81.36 \pm 0.01$ | $0.8882 \pm 0.0001$ | $5.77 \pm 0.01$ |
| | Transformer | 1,941 | 2,479,761 | $81.64 \pm 0.03$ | $0.8913 \pm 0.0001$ | $5.98 \pm 0.03$ |
| 2 | SAL-T | 6,171 | 2,095,761 | $\mathbf{81.77 \pm 0.02}$ | $\mathbf{0.8924 \pm 0.0004}$ | $\mathbf{6.07 \pm 0.01}$ |
| | Linformer | 13,129 | 1,087,761 | $81.60 \pm 0.08$ | $0.8906 \pm 0.0006$ | $5.94 \pm 0.05$ |
| | Transformer | 3,529 | 4,942,161 | $82.09 \pm 0.03$ | $0.8957 \pm 0.0004$ | $6.32 \pm 0.02$ |

To assess the robustness of SAL-T, we evaluate its performance alongside baseline models on two additional high-energy physics datasets commonly used in the literature. Given that these datasets are significantly larger than `hls4ml`, we report results for both one-layer and two-layer attention models, whereas only single-layer models are considered for `hls4ml`. As shown in Tables 3 and 16, while the performance of single-layer SAL-T and Linformer models is within statistical uncertainty, SAL-T outperforms linformer with two attention layers, indicating that SAL-T scales more effectively with model capacity and is better suited for larger datasets.

Notably, on the Quark Gluon dataset (Table 16), the two-layer SAL-T surpasses the single-layer Transformer in classification performance while also offering substantial gains in computational efficiency, highlighting the scalability and expressiveness of SAL-T in more complex scenarios.

**Ablation Study** We find that using the partition and convolution individually increases the performance of SAL-T when compared the base linformer. Both partitioning and convolution increases performance of the base linformer for longer, more complex jets, as seen in Table 5. Individually, SAL-T without partition increases performance when there are 76–114 particles in a jet. SAL-T without the convolution (only using partition) increases overall accuracy, along with accuracy for 76–114 particles to above that of linformer (see Figure 3). Furthermore, combining partition and convolution to make the full LPP-MHA layer further increases the performance. This demonstrates that both novel components of LPP-MHA individually learn spatial information about the jets.

Table 5: Ablation study on SAL-T components. Reported metrics include overall accuracy and accuracy in bins 2 and 3 of number of particles in a jet reported in Figure 3, as well as background rejection at 0.8 signal efficiency.

| Model | # Params | Accuracy [%] | Bin 2 (38-75) [%] | Bin 3 (76-114) [%] | 1/FPR@0.8TPR |
|---|---|---|---|---|---|
| SAL-T ($p_\mathrm{T}$ sorted) | 3,264 | $78.82 \pm 0.46$ | $78.02 \pm 0.50$ | $76.24 \pm 4.42$ | $23.45 \pm 2.12$ |
| SAL-T (no partition) | 6,848 | $81.02 \pm 0.05$ | $80.40 \pm 0.05$ | $81.15 \pm 0.69$ | $27.68 \pm 0.67$ |
| SAL-T (no convolution) | 3,225 | $81.09 \pm 0.10$ | $80.55 \pm 0.08$ | $81.60 \pm 0.93$ | $17.90 \pm 2.64$ |
| Linformer ($p_\mathrm{T}$ sorted) | 6,809 | $79.90 \pm 0.06$ | $79.45 \pm 0.13$ | $80.80 \pm 0.42$ | $28.06 \pm 0.58$ |
| Linformer ($k_\mathrm{T}$ sorted) | 6,809 | $81.00 \pm 0.08$ | $80.45 \pm 0.04$ | $80.41 \pm 0.08$ | $38.41 \pm 0.54$ |
| SAL-T ($k_\mathrm{T}$ sorted) | 3,264 | $\mathbf{81.18 \pm 0.03}$ | $\mathbf{80.60 \pm 0.06}$ | $\mathbf{81.60 \pm 0.20}$ | $\mathbf{40.78 \pm 0.57}$ |

**ModelNet10 Results.** To further highlight the applicability of SAL-T to machine learning tasks beyond particle physics, we benchmarked on the well-known ModelNet10 dataset. In this setting, each point cloud is represented by a fixed set of 1024 points, and the task is to classify objects into ten distinct categories. We sorted the input points by Morton codes to preserve spatial locality in a way analogous to $k_T$ sorting for jets, and found that this ordering is crucial: without it, both SAL-T and Linformer show a marked drop in performance. We use the exact same model as used for the hls4ml dataset. (Table 6) With sorting, SAL-T substantially outperforms Linformer in accuracy and AUC, while being more efficient than a full Transformer. As expected, the full Transformer achieves the highest accuracy, but at the cost of quadratic compute and memory growth with sequence length. In contrast, SAL-T offers a favorable trade-off between accuracy and efficiency, scaling linearly while retaining competitive discriminative power. These results show that the spatially aware design of SAL-T can leverage input orderings beyond those used in particle physics, making it a generally useful approach for point cloud classification tasks.

Table 6: ModelNet10 performance comparison of Transformer, Linformer, and SAL-T. Results are reported as mean $\pm$ standard deviation.

| Model | FLOPs | Peak GPU Mem [MB] | Accuracy [%] |
|---|---|---|---|
| Transformer | $95,683,468 \pm 0$ | $6,221.1 \pm 8.3$ | $82.56 \pm 3.36$ |
| Linformer | $3,769,228 \pm 0$ | $397.8 \pm 27.1$ | $77.86 \pm 1.16$ |
| SAL-T | $5,047,180 \pm 0$ | $454.6 \pm 3.1$ | $80.10 \pm 1.11$ |
| Linformer (unsorted) | $3,769,228 \pm 0$ | $397.8 \pm 27.1$ | $60.72 \pm 1.97$ |
| SAL-T (unsorted) | $5,047,180 \pm 0$ | $454.6 \pm 3.1$ | $68.79 \pm 1.16$ |

## 5 SUMMARY AND OUTLOOK

In this work, we have presented SAL-T, a spatially aware linear transformer that incorporates partitioned attention and lightweight convolution to encode locality in jet constituents. Across multiple jet-tagging benchmarks, SAL-T consistently outperforms linformer in accuracy, area under the receiver operating characteristic curve, and background rejection, achieves performance comparable to full-attention transformers, and maintains linformer-level FLOPs and inference latency, resulting in significantly improved efficiency over full transformers. These results demonstrate that integrating physics-informed locality into low-rank attention mechanisms can yield substantial gains in performance for real-time collider data analysis.

**Limitations and Future Directions** While SAL-T exhibits robust performance and notable efficiency improvements on standard jet tagging benchmarks, several limitations persist. First, our evaluation has been constrained to the `hls4ml`, Top Tagging, and Quark Gluon datasets. Demonstrating comparable performance on real CMS or ATLAS trigger data and on other point cloud tasks would enhance confidence in its general applicability. Second, the current spatially aware partitioning employs a single choice of $p$ and fixed convolution filter sizes with heights of 1, 3, and 5. The incorporation of adaptive partition ranks or the ability to learn multi-scale convolution kernels could lead to more comprehensive local feature representations. Third, all experiments utilize a maximum of two LPP-MHA layers; the employment of deeper SAL-T stacks might offer a better capture of jets' hierarchical structure, albeit potentially compromising the latency benefits we have observed. Lastly, our latency benchmarks were conducted on a desktop-class GPU (NVIDIA GeForce GTX 1080 Ti). Integrating SAL-T into FPGA-based trigger systems could present additional challenges in adhering to stringent hardware constraints.

Looking forward, we plan to pursue several avenues to address these limitations. We will integrate spatially aware partitioning and convolution with the pairwise feature attention bias of particle transformer (3), merging its multi-scale attention scheme with our physics-informed projections to achieve combined benefits. For our future implementations, we plan to incorporate a sorting techniques based on the clustering sequence history of the particle jets (21; 38), for a more physics informed partitioning. We aim to explore dynamic partitioning, enabling the model to adjust partition sizes or counts dynamically in response to particle multiplicity, potentially enhancing efficiency for jets with varying complexity. Moreover, hardware-aware optimization by tailoring SAL-T kernels for FPGAs or low-power inference engines is intended to meet trigger latency requirements in the microseconds range. Finally, we intend to deploy SAL-T across a broader spectrum of collider-based point-cloud tasks, such as jet reconstruction and anomaly detection, to evaluate its generalizability beyond jet tagging.

## ETHICS STATEMENT

This work introduces SAL-T, a transformer-based architecture designed for efficient particle jet classification in high-energy physics. The datasets used in this study are widely adopted public benchmarks: the `hls4ml` dataset (24; 25), the Top Tagging dataset (27; 28), the Quark–Gluon dataset (29; 30), and ModelNet10 (31). These datasets contain only simulated or synthetic data, and thus do not involve human subjects, personally identifiable information, or sensitive attributes. We therefore do not anticipate risks related to privacy, security, or discrimination.

Our primary application domain is high-energy physics, where more efficient ML models can reduce computational costs for large-scale experiments, thereby decreasing energy usage and improving sustainability of data processing at facilities such as the LHC. We acknowledge that any advances in efficient machine learning can be repurposed for other domains; however, this work does not target or develop applications with direct societal risks, such as surveillance or military use. All experiments were conducted in compliance with open-source licensing terms, and we release our code in anonymized form to promote transparency and reproducibility.

We believe this work aligns with the ICLR Code of Ethics by upholding principles of research integrity, fairness, and responsible dissemination.

## REPRODUCIBILITY STATEMENT

We have taken several steps to ensure the reproducibility of our results. All datasets used in this work are publicly available benchmarks, and we describe their preprocessing and train/validation/test splits in Section 3. The full model architecture, including hyperparameters and training details, is provided in Section 2 and Section 3. We report results averaged over multiple runs with standard deviations where applicable. An anonymized implementation of SAL-T, along with training and evaluation scripts, is included in the abstract to facilitate verification and extension of our work.

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

# SUPPLEMENTARY MATERIAL

## LLM USAGE

Large Language Models (LLMs) were used as a general purpose writing and editing assistant in the preparation of this manuscript. Specifically, LLMs helped with phrasing improvements, grammar checks, and formatting of LaTeX tables and sections. All research ideas, experiments, analyses, and conclusions were conceived and carried out by the authors. The authors have carefully verified the accuracy of all content and take full responsibility for the final text.

## HYPERPARAMETER STUDIES

We perform ablation studies on two key hyperparameters of SAL-T: the number of partitions and the convolution filter sizes. Table 7 shows that for 150 particles per jet, increasing the number of partitions initially improves both efficiency and performance, but beyond four partitions the computational cost grows substantially without additional performance gains. We therefore select four partitions as the default setting. Table 8 compares different convolution filter configurations, where the default setting of $\{1, 3, 5\}$ achieves the best overall trade-off between accuracy, AUC, and efficiency.

Table 7: Effect of partition number on FLOPs, inference time, peak GPU memory, and classification performance.

| Partitions | FLOPs | Inference [$\mu s$] | Peak Mem [MB] | Accuracy [%] | ROC AUC | 1/FPR@0.8TPR |
|---|---|---|---|---|---|---|
| 1 | 527,518 | 24.71 ± 0.88 | 227.9 | 80.66 ± 0.03 | 0.9573 | 35.24 ± 0.65 |
| 2 | 576,718 | 27.00 ± 0.23 | 228.4 | 80.95 ± 0.13 | 0.9583 | 37.91 ± 1.31 |
| **4** | **739,918** | **27.51 ± 0.35** | **303.4** | **81.18 ± 0.03** | **0.9593** | **40.78 ± 0.57** |
| 8 | 1,325,518 | 34.58 ± 0.74 | 530.4 | 81.10 ± 0.13 | 0.9591 | 39.75 ± 1.21 |
| 16 | 3,533,518 | 56.91 ± 0.89 | 981.4 | 80.97 ± 0.19 | 0.9583 | 38.48 ± 1.80 |

Table 8: Effect of convolution filter sizes on FLOPs, inference time, peak GPU memory, and classification performance.

| Filter sizes | FLOPs | Inference [$\mu s$] | Peak Mem [MB] | Accuracy [%] | ROC AUC | 1/FPR@0.8TPR |
|---|---|---|---|---|---|---|
| **1 3 5** | **739,918** | **27.51 ± 0.35** | **303.4** | **81.18 ± 0.03** | **0.9593** | **40.78 ± 0.57** |
| 1 5 7 | 816,718 | 29.16 ± 0.45 | 303.4 | 81.00 ± 0.06 | 0.9588 | 38.70 ± 1.14 |
| 1 3 5 7 | 879,118 | 30.36 ± 0.66 | 378.4 | 81.10 ± 0.10 | 0.9589 | 39.25 ± 0.70 |
| 1 3 5 7 9 | 1,056,718 | 32.96 ± 0.36 | 453.4 | 81.02 ± 0.12 | 0.9587 | 38.63 ± 1.13 |
| 3 3 3 | 739,918 | 29.66 ± 0.14 | 303.4 | 80.96 ± 0.03 | 0.9585 | 37.92 ± 0.70 |
| 5 5 5 | 855,118 | 29.34 ± 0.54 | 303.4 | 80.97 ± 0.11 | 0.9585 | 38.15 ± 1.29 |
| 7 7 7 | 970,318 | 30.62 ± 0.45 | 303.4 | 80.98 ± 0.12 | 0.9587 | 38.58 ± 1.27 |
| 3 5 7 | 855,118 | 29.53 ± 0.38 | 303.4 | 81.07 ± 0.05 | 0.9588 | 39.13 ± 0.40 |
| 3 5 7 9 | 1,032,718 | 31.15 ± 0.27 | 378.4 | 81.09 ± 0.03 | 0.9589 | 39.53 ± 0.22 |

## FURTHER ABLATION STUDIES

In addition to the above studies, we also conduct a full ablation study of SAL-T components on `hls4ml` dataset constrained to a maximum of 16, 32, and 150 particles to match the results in (26) We use the following ablations to the full SAL-T model: Only Partitioning the value matrix, only partitioning the key matrix, only using one set of projections for both the key and value matrix (Share EF), without convolution, and without partitioning. We repeat these experiments for inputs sorted by all $p_T$, $\Delta R$, and $k_T$. We find that $k_T$ sorting of inputs consistently performs better than other sorting methods regardless of input length, as shown in Tables 9 13 14. Additionally, we find that SAL-T is most effective at tagging on longer seqeuences.

Individually, when benchmarking on 150 particles, we find that partitioning the value matrix is especially important to achieve strong accuracy and rejection. In Table 13, full SAL-T, SAL-T

where only the value matrix is partitioned, SAL-T where the value and key matrix share partitioning weights perform the best when compared to other ablations.

In Tables 11 12 and 14 we demonstrate that partitioning reduces parameter count compared to linformer, and convolution with 3 filters adds a very small number of parameters when compared to linformer. We show that SAL-T and linformer uses much less GPU memory at its peak.

SOTA BASELINE UNDER TINY-MODEL CONSTRAINTS

Since SAL-T was designed to be implemented onto FPGAs, we focus on tiny machine learning models. The performance of tiny machine learning models for particle physics is currently an **under-explored area**. We apply the current state-of-the-art Particle Transformer (ParT) and perform an apples-to-apples comparison with SAL-T by shrinking ParT to the size of our tiny models and running benchmarks. As shown in Table 10, ParT improves performance compared to the native transformer and SAL-T, but drastically increases the resource usage due to the convolution layers needed to process the pairwise mask, and the computation of the pairwise features.

ATTENTION MATRICES

We plot attention matrices to demonstrate that convolution and partitioning allows for a more interpretable model. For instance, in figures 4 5 6, we are able to identify which range of particles SAL-T is attending. On the other hand, since the vanilla linformer has projections that are multiples of all particles, the projections are much less interpretable. The convolution in SAL-T smooths the attention in the attention maps, visualizing how SAL-T learns local connections in attention. We take the attention matrix of all particles after running inference on the test dataset, and sum the attention values to the first partition. Then, we divide it by the sum of all the attention values in the matrix. Interestingly, we find that head 2 of SAL-T almost always attends to partition 0 as seen in figure 7. This only occurs in SAL-T full, and does not occur in linformer.

BATCH SIZE SCHEDULING VS LEARNING RATE DECAY

We trained all models on a phased batch size scheduling training scheme. We doubled the batch size starting from 128 all the way to 4096 every 200 epochs, with early stopping for each batch size if the validation loss does not decrease for 40 epochs. We also tested learning rate decay, where we start at 0.001 learning rate using adam optimizer, and decreased it by half whenever the validation loss plateaued for 80 epochs. We find that the batch size scheduling outperforms the learning rate decay as seen in figure 8.

Table 9: Performance metrics for hls4ml with at most 16 particles per jet

| Model | Sort | Test Accuracy [%] | ROC AUC | Avg 1/FPR |
|---|---|---|---|---|
| SAL-T Partition Value Only | deltaR | 72.12 ± 0.06 | 0.92 ± 0.00 | 10.60 ± 0.06 |
| SAL-T | deltaR | 72.28 ± 0.25 | 0.92 ± 0.00 | 10.82 ± 0.31 |
| SAL-T No Conv Partition Value Only | deltaR | 72.30 ± 0.40 | 0.92 ± 0.00 | 10.81 ± 0.40 |
| SAL-T Share EF | pt | 72.32 ± 0.09 | 0.92 ± 0.00 | 10.84 ± 0.08 |
| SAL-T No Conv | pt | 72.35 ± 0.19 | 0.92 ± 0.00 | 10.85 ± 0.20 |
| SAL-T Share EF | deltaR | 72.35 ± 0.26 | 0.92 ± 0.00 | 10.86 ± 0.29 |
| SAL-T | pt | 72.39 ± 0.04 | 0.92 ± 0.00 | 10.91 ± 0.02 |
| SAL-T No Conv | deltaR | 72.39 ± 0.04 | 0.92 ± 0.00 | 10.89 ± 0.05 |
| SAL-T Partition Value Only | pt | 72.54 ± 0.07 | 0.92 ± 0.00 | 11.09 ± 0.17 |
| SAL-T No Conv Partition Value Only | pt | 72.61 ± 0.07 | 0.92 ± 0.00 | 11.19 ± 0.06 |
| SAL-T No Conv Partition Key Only | deltaR | 72.75 ± 0.05 | 0.93 ± 0.00 | 11.35 ± 0.07 |
| SAL-T No Partition | deltaR | 72.78 ± 0.09 | 0.93 ± 0.00 | 11.38 ± 0.14 |
| SAL-T No Conv Partition Key Only | pt | 72.84 ± 0.05 | 0.93 ± 0.00 | 11.39 ± 0.10 |
| SAL-T No Partition | pt | 72.85 ± 0.12 | 0.93 ± 0.00 | 11.45 ± 0.13 |
| Linformer | deltaR | 72.92 ± 0.10 | 0.93 ± 0.00 | 11.47 ± 0.08 |
| SAL-T Partition Key Only | pt | 72.94 ± 0.06 | 0.93 ± 0.00 | 11.52 ± 0.06 |
| Linformer | pt | 72.94 ± 0.18 | 0.93 ± 0.00 | 11.52 ± 0.13 |
| SAL-T Partition Key Only | deltaR | 72.95 ± 0.13 | 0.93 ± 0.00 | 11.58 ± 0.13 |
| SAL-T | kt | 73.07 ± 0.27 | 0.93 ± 0.00 | 11.72 ± 0.24 |
| SAL-T No Conv | kt | 73.12 ± 0.12 | 0.93 ± 0.00 | 11.76 ± 0.06 |
| SAL-T No Conv Partition Value Only | kt | 73.22 ± 0.09 | 0.93 ± 0.00 | 11.87 ± 0.14 |
| SAL-T No Conv Partition Key Only | kt | 73.24 ± 0.09 | 0.93 ± 0.00 | 11.93 ± 0.13 |
| SAL-T Partition Value Only | kt | 73.29 ± 0.06 | 0.93 ± 0.00 | 12.00 ± 0.02 |
| SAL-T Partition Key Only | kt | 73.35 ± 0.06 | 0.93 ± 0.00 | 12.03 ± 0.09 |
| SAL-T Share EF | kt | 73.40 ± 0.20 | 0.93 ± 0.00 | 12.09 ± 0.24 |
| SAL-T No Partition | kt | 73.52 ± 0.03 | 0.93 ± 0.00 | 12.27 ± 0.04 |
| Linformer | kt | 73.58 ± 0.08 | 0.93 ± 0.00 | 12.31 ± 0.15 |
| Transformer | pt | 73.60 ± 0.13 | 0.93 ± 0.00 | 12.48 ± 0.13 |

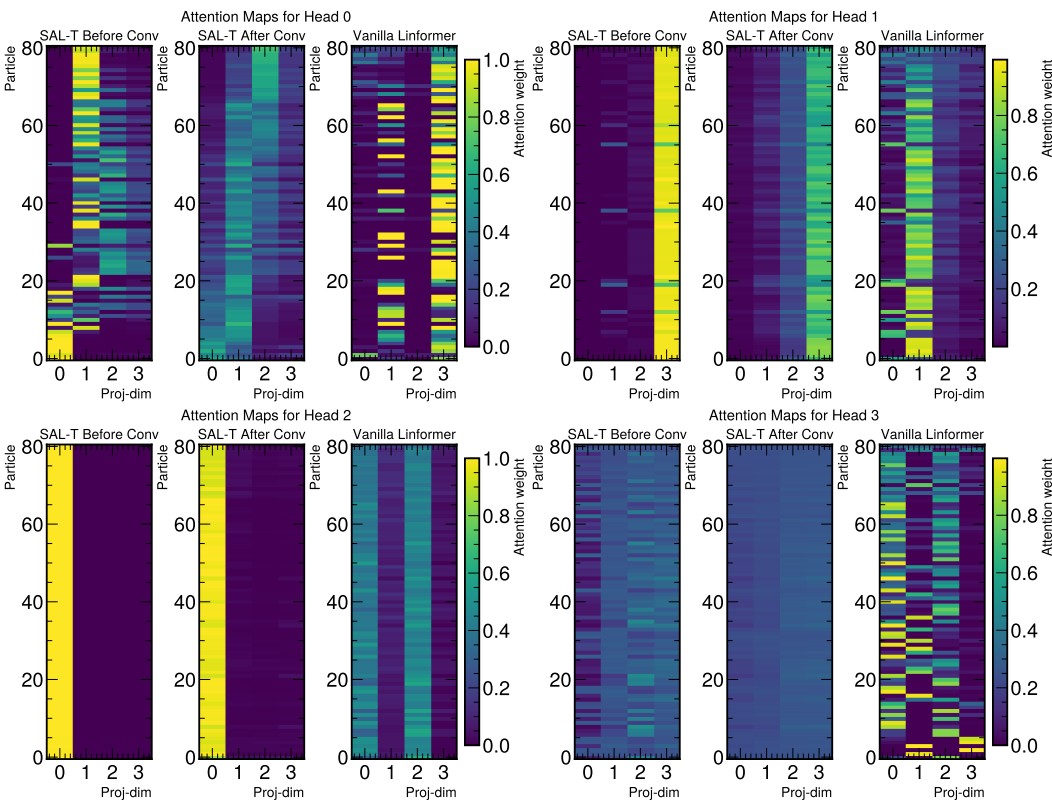

Figure 4: Attention matrices for a top quark jet with 81 particles. Each trio of attention plots represent a separate head of each respective model. The convolutional layer on top of the attention smooths the attention values of SAL-T, demonstrating how convolution helps SAL-T leverage immediate neigbhors. Notice in the top right and bottom left attention plots, SAL-T attention focuses on only one partition before convolution, signifying that SAL-T understands the partitions that are important to the jet.

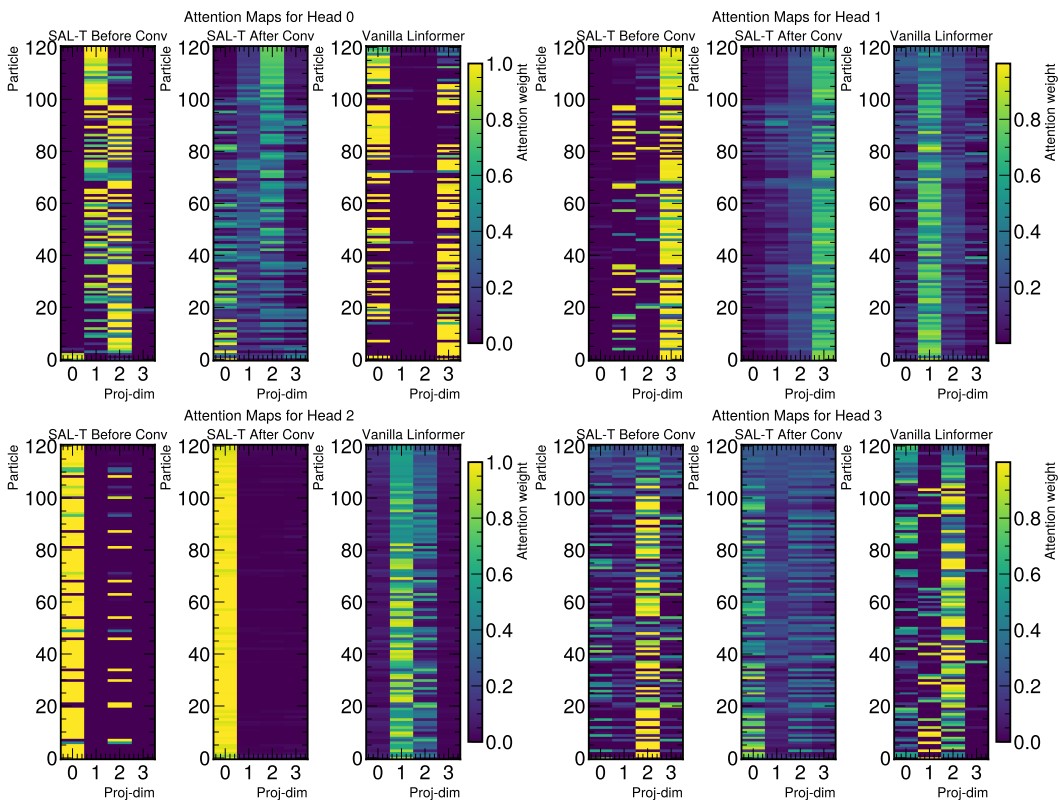

Figure 5: Attention matrices for a top quark jet with 121 particles. Each trio of attention matrices represents a separate head of the respective models. The convolutional layer on top of the attention smooths the attention values of SAL-T, demonstrating how convolution helps SAL-T leverage immediate neighbors.

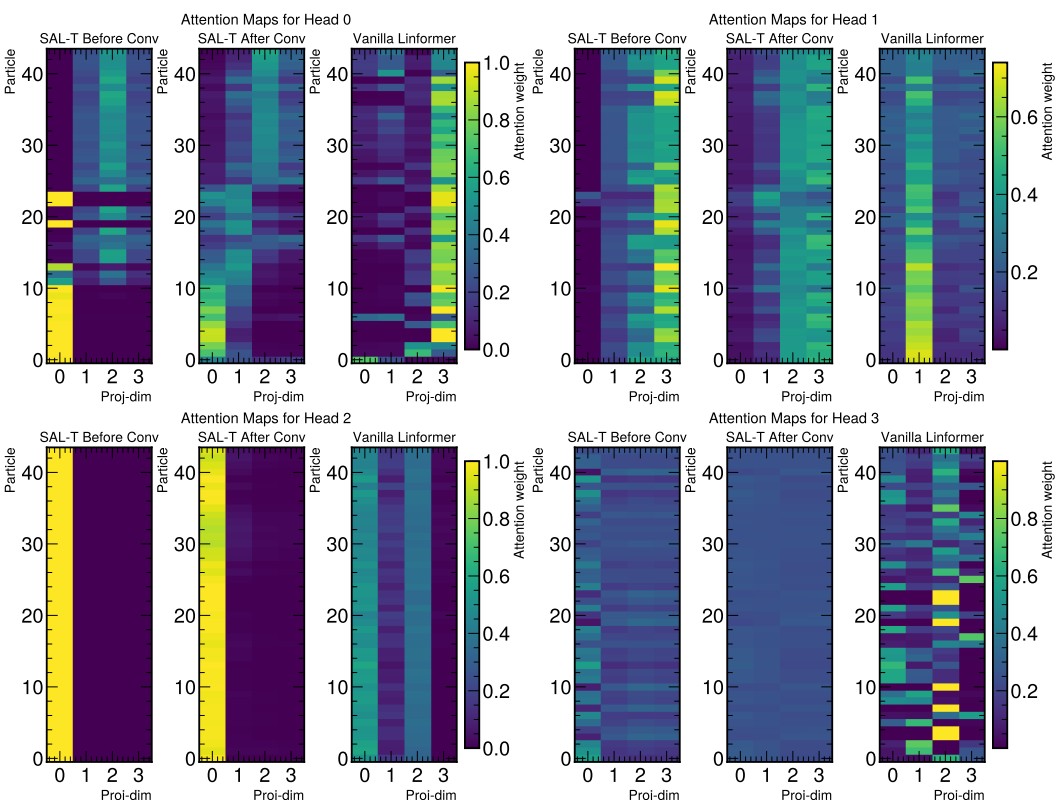

Figure 6: Attention matrices for a top quark jet with 44 particles. Each trio of attention matrices represents a separate head of the respective models. The convolutional layer on top of the attention smooths the attention values of SAL-T, demonstrating how convolution helps SAL-T leverage immediate neighbors.

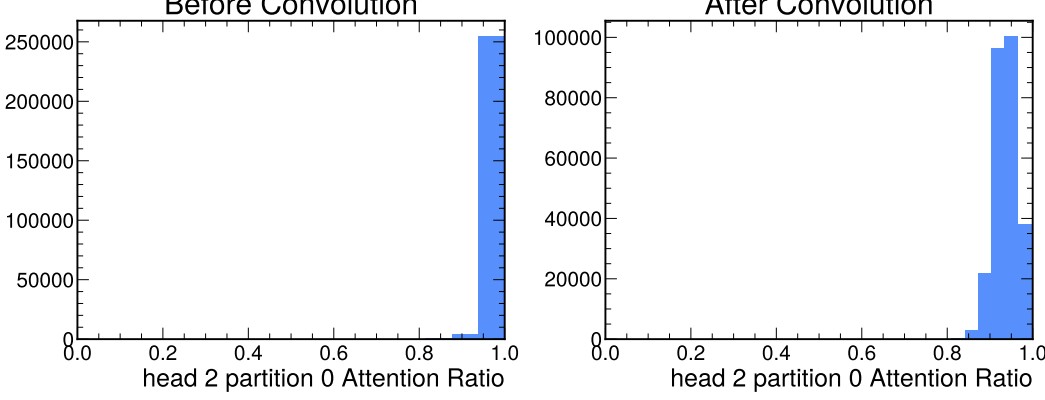

Figure 7: histogram of ratio of attention to partition 0 in head 2 throughout all particles in the test set. We show that before convolution, head 2 almost always attends to partition 0.

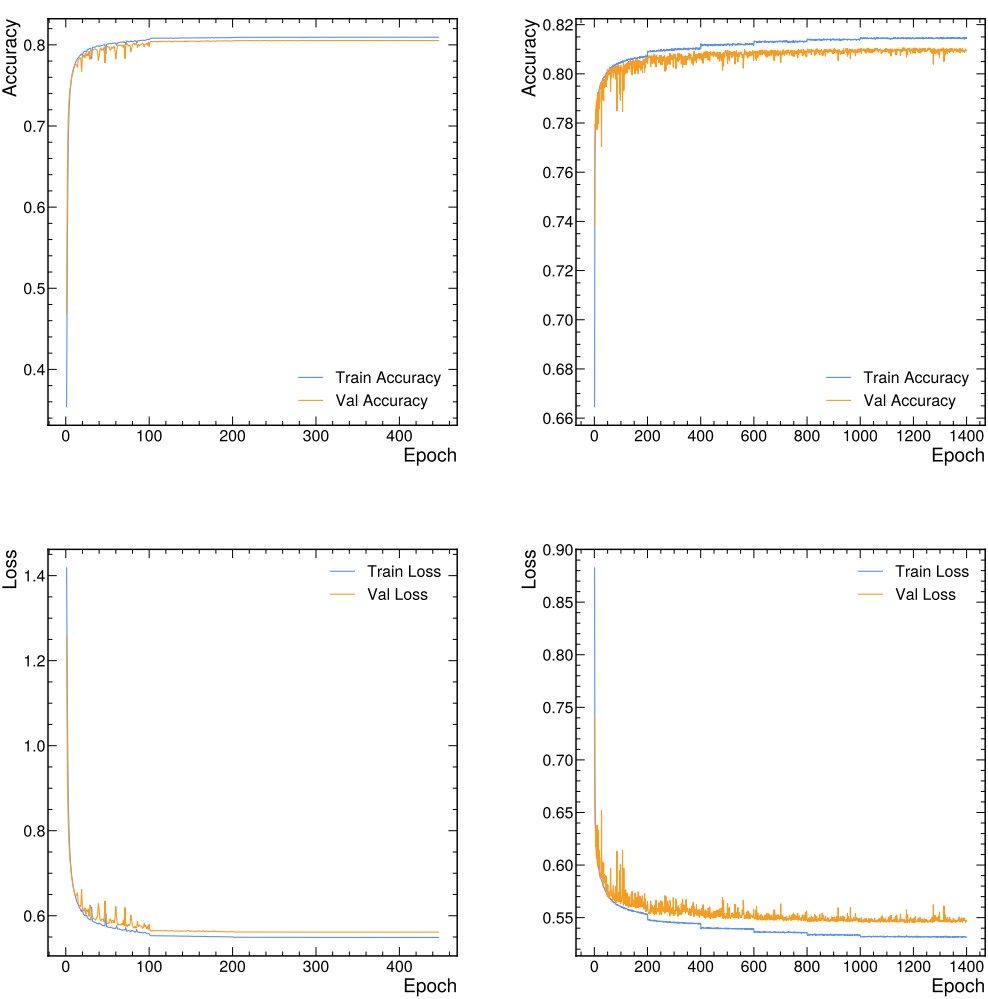

Figure 8: Training/validation accuracy and loss curves for SAL-T with learning rate decay vs batch size scheduling. *(Top Left)* Accuracy curve for learning rate decay. *(Top Right)* Accuracy Curve for batch size scheduling. *(Bottom Left)* Loss curve for learning rate decay. *(Bottom Right)* Loss curve for batch size scheduling.

Table 10: Comparison of SAL-T with a reduced Particle Transformer and a native transformer.

| Model | FLOPs | Inference [$\mu$s] | Peak Mem [MB] | Accuracy [%] | ROC AUC | 1/FPR@0.8TPR |
|---|---|---|---|---|---|---|
| 1-layer-ParT | 8,217,868 | 211.20 $\pm$ 0.84 | OOM | 81.42 $\pm$ 0.19 | 0.9599 | 43.16 $\pm$ 1.67 |
| **SAL-T** | **739,918** | **26.78 $\pm$ 0.13** | **303.4** | **81.18 $\pm$ 0.03** | **0.9593** | **40.78 $\pm$ 0.57** |
| transformer | 2,479,918 | 29.61 $\pm$ 0.12 | 4,357.1 | 81.27 $\pm$ 0.08 | 0.9589 | 42.02 $\pm$ 0.71 |

Table 11: Efficiency metrics for 16-particle HLS4ML datset

| Model | Sort | Training Time [h] | Params | GPU Peak [MB] | FLOPs | Inference Time [µs] |
|---|---|---|---|---|---|---|
| SAL-T Partition Value Only | deltaR | 3.84 $\pm$ 0.09 | 2368 | 37.0 $\pm$ 0.0 | 79566 | 15.45 $\pm$ 0.28 |
| SAL-T | deltaR | 4.21 $\pm$ 0.25 | 2176 | 38.3 $\pm$ 0.0 | 79566 | 15.78 $\pm$ 0.14 |
| SAL-T No Conv Partition Value Only | deltaR | 2.93 $\pm$ 0.50 | 2329 | 29.0 $\pm$ 0.0 | 59598 | 15.22 $\pm$ 0.16 |
| SAL-T Share EF | pt | 3.53 $\pm$ 0.36 | 2112 | 38.3 $\pm$ 0.0 | 79566 | 15.78 $\pm$ 0.19 |
| SAL-T No Conv | pt | 3.00 $\pm$ 0.25 | 2137 | 27.3 $\pm$ 0.0 | 59598 | 15.21 $\pm$ 0.26 |
| SAL-T Share EF | deltaR | 3.53 $\pm$ 0.29 | 2112 | 38.3 $\pm$ 0.0 | 79566 | 15.88 $\pm$ 0.18 |
| SAL-T | pt | 4.09 $\pm$ 0.35 | 2176 | 38.3 $\pm$ 0.0 | 79566 | 15.53 $\pm$ 0.07 |
| SAL-T No Conv | deltaR | 3.05 $\pm$ 0.23 | 2137 | 27.3 $\pm$ 0.0 | 59598 | 15.05 $\pm$ 0.06 |
| SAL-T Partition Value Only | pt | 3.73 $\pm$ 0.15 | 2368 | 37.0 $\pm$ 0.0 | 79566 | 15.54 $\pm$ 0.14 |
| SAL-T No Conv Partition Value Only | pt | 2.91 $\pm$ 0.47 | 2329 | 29.0 $\pm$ 0.0 | 59598 | 15.30 $\pm$ 0.20 |
| SAL-T No Conv Partition Key Only | deltaR | 2.90 $\pm$ 0.43 | 2329 | 29.0 $\pm$ 0.0 | 59598 | 15.26 $\pm$ 0.16 |
| SAL-T No Partition | deltaR | 3.21 $\pm$ 0.16 | 2560 | 37.1 $\pm$ 0.0 | 79566 | 15.43 $\pm$ 0.32 |
| SAL-T No Conv Partition Key Only | pt | 2.86 $\pm$ 0.42 | 2329 | 29.0 $\pm$ 0.0 | 59598 | 15.12 $\pm$ 0.35 |
| SAL-T No Partition | pt | 3.20 $\pm$ 0.15 | 2560 | 37.1 $\pm$ 0.0 | 79566 | 15.55 $\pm$ 0.12 |
| Linformer | deltaR | 2.75 $\pm$ 0.12 | 2521 | 29.0 $\pm$ 0.0 | 59598 | 15.00 $\pm$ 0.16 |
| SAL-T Partition Key Only | pt | 3.27 $\pm$ 0.37 | 2368 | 41.5 $\pm$ 0.0 | 79566 | 15.62 $\pm$ 0.18 |
| Linformer | pt | 2.76 $\pm$ 0.17 | 2521 | 29.0 $\pm$ 0.0 | 59598 | 15.07 $\pm$ 0.28 |
| SAL-T Partition Key Only | deltaR | 3.39 $\pm$ 0.37 | 2368 | 41.5 $\pm$ 0.0 | 79566 | 15.71 $\pm$ 0.24 |
| SAL-T | kt | 4.07 $\pm$ 0.26 | 2176 | 38.3 $\pm$ 0.0 | 79566 | 15.70 $\pm$ 0.33 |
| SAL-T No Conv | kt | 3.13 $\pm$ 0.31 | 2137 | 27.3 $\pm$ 0.0 | 59598 | 15.14 $\pm$ 0.16 |
| SAL-T No Conv Partition Value Only | kt | 2.97 $\pm$ 0.48 | 2329 | 29.0 $\pm$ 0.0 | 59598 | 15.33 $\pm$ 0.18 |
| SAL-T No Conv Partition Key Only | kt | 2.86 $\pm$ 0.44 | 2329 | 29.0 $\pm$ 0.0 | 59598 | 15.16 $\pm$ 0.28 |
| SAL-T Partition Value Only | kt | 3.83 $\pm$ 0.06 | 2368 | 37.0 $\pm$ 0.0 | 79566 | 15.69 $\pm$ 0.16 |
| SAL-T Partition Key Only | kt | 3.32 $\pm$ 0.45 | 2368 | 41.5 $\pm$ 0.0 | 79566 | 15.43 $\pm$ 0.11 |
| SAL-T Share EF | kt | 3.58 $\pm$ 0.38 | 2112 | 38.3 $\pm$ 0.0 | 79566 | 15.61 $\pm$ 0.26 |
| SAL-T No Partition | kt | 3.26 $\pm$ 0.24 | 2560 | 37.1 $\pm$ 0.0 | 79566 | 15.57 $\pm$ 0.10 |
| Linformer | kt | 2.69 $\pm$ 0.07 | 2521 | 29.0 $\pm$ 0.0 | 59598 | 15.09 $\pm$ 0.02 |
| Transformer | pt | 1.98 $\pm$ 0.34 | 2009 | 62.8 $\pm$ 0.0 | 76494 | 15.21 $\pm$ 0.18 |

Table 12: Performance metrics for 32-particle HLS4ML dataset

| Model | Sort | Test Accuracy [%] | ROC AUC | Avg 1/FPR |
|---|---|---|---|---|
| SAL-T Share EF | deltaR | 76.97 ± 0.20 | 0.94 ± 0.00 | 17.89 ± 0.68 |
| SAL-T | deltaR | 77.06 ± 0.72 | 0.94 ± 0.00 | 18.22 ± 2.04 |
| SAL-T No Conv | deltaR | 77.16 ± 0.35 | 0.94 ± 0.00 | 18.22 ± 1.04 |
| SAL-T No Conv Partition Value Only | deltaR | 77.36 ± 0.63 | 0.94 ± 0.00 | 19.05 ± 1.94 |
| SAL-T No Conv Partition Value Only | pt | 77.62 ± 0.26 | 0.95 ± 0.00 | 19.73 ± 0.93 |
| SAL-T | pt | 77.63 ± 0.12 | 0.95 ± 0.00 | 19.89 ± 0.61 |
| SAL-T Share EF | pt | 77.63 ± 0.22 | 0.95 ± 0.00 | 19.86 ± 0.58 |
| SAL-T Partition Value Only | pt | 77.66 ± 0.29 | 0.95 ± 0.00 | 20.11 ± 0.98 |
| SAL-T No Conv | pt | 77.69 ± 0.07 | 0.95 ± 0.00 | 20.11 ± 0.15 |
| SAL-T Partition Value Only | deltaR | 77.74 ± 0.39 | 0.95 ± 0.00 | 20.32 ± 1.27 |
| SAL-T No Conv Partition Key Only | pt | 77.97 ± 0.40 | 0.95 ± 0.00 | 21.17 ± 1.35 |
| SAL-T No Conv Partition Key Only | deltaR | 78.08 ± 0.14 | 0.95 ± 0.00 | 21.44 ± 0.41 |
| SAL-T Partition Key Only | deltaR | 78.21 ± 0.08 | 0.95 ± 0.00 | 21.81 ± 0.24 |
| Linformer | pt | 78.26 ± 0.19 | 0.95 ± 0.00 | 22.09 ± 0.60 |
| Linformer | deltaR | 78.32 ± 0.26 | 0.95 ± 0.00 | 22.39 ± 0.98 |
| SAL-T No Partition | deltaR | 78.34 ± 0.17 | 0.95 ± 0.00 | 22.45 ± 0.60 |
| SAL-T No Partition | pt | 78.39 ± 0.04 | 0.95 ± 0.00 | 22.44 ± 0.33 |
| SAL-T Partition Key Only | pt | 78.39 ± 0.16 | 0.95 ± 0.00 | 22.75 ± 0.49 |
| Transformer | deltaR | 78.52 ± 1.36 | 0.95 ± 0.01 | 25.46 ± 4.67 |
| SAL-T No Conv Partition Value Only | kt | 78.79 ± 0.18 | 0.95 ± 0.00 | 25.29 ± 1.00 |
| SAL-T No Conv | kt | 78.81 ± 0.04 | 0.95 ± 0.00 | 25.38 ± 0.21 |
| SAL-T No Conv Partition Key Only | kt | 78.88 ± 0.08 | 0.95 ± 0.00 | 25.79 ± 0.27 |
| SAL-T Share EF | kt | 78.90 ± 0.15 | 0.95 ± 0.00 | 25.66 ± 0.73 |
| SAL-T Partition Key Only | kt | 78.92 ± 0.16 | 0.95 ± 0.00 | 25.55 ± 0.46 |
| SAL-T Partition Value Only | kt | 79.02 ± 0.06 | 0.95 ± 0.00 | 26.23 ± 0.62 |
| SAL-T | kt | 79.02 ± 0.12 | 0.95 ± 0.00 | 25.98 ± 0.97 |
| SAL-T No Partition | kt | 79.06 ± 0.12 | 0.95 ± 0.00 | 26.67 ± 0.54 |
| Transformer | kt | 79.08 ± 0.04 | 0.95 ± 0.00 | 27.29 ± 0.12 |
| Transformer | pt | 79.10 ± 0.12 | 0.95 ± 0.00 | 27.33 ± 0.47 |
| Linformer | kt | 79.13 ± 0.10 | 0.95 ± 0.00 | 26.75 ± 0.50 |

Table 13: Efficiency metrics for 32-particle HLS4ML dataset

| Model | Sort | Training Time [h] | Params | GPU Peak [MB] | FLOPs | Inference Time [μs] |
|---|---|---|---|---|---|---|
| SAL-T Share EF | deltaR | 3.91 ± 0.33 | 2176 | 65.5 ± 0.0 | 158414 | 17.26 ± 0.18 |
| SAL-T | deltaR | 4.44 ± 0.20 | 2304 | 65.5 ± 0.0 | 158414 | 17.19 ± 0.24 |
| SAL-T No Conv | deltaR | 3.31 ± 0.32 | 2265 | 53.0 ± 0.0 | 118478 | 16.36 ± 0.05 |
| SAL-T No Conv Partition Value Only | deltaR | 2.97 ± 0.34 | 2649 | 55.5 ± 0.0 | 118478 | 16.18 ± 0.10 |
| SAL-T No Conv Partition Value Only | pt | 2.98 ± 0.40 | 2649 | 55.5 ± 0.0 | 118478 | 16.19 ± 0.19 |
| SAL-T | pt | 4.42 ± 0.23 | 2304 | 65.5 ± 0.0 | 158414 | 17.22 ± 0.35 |
| SAL-T Share EF | pt | 3.91 ± 0.28 | 2176 | 65.5 ± 0.0 | 158414 | 17.15 ± 0.13 |
| SAL-T Partition Value Only | pt | 4.20 ± 0.06 | 2688 | 64.1 ± 0.0 | 158414 | 17.05 ± 0.30 |
| SAL-T No Conv | pt | 3.34 ± 0.39 | 2265 | 53.0 ± 0.0 | 118478 | 16.04 ± 0.23 |
| SAL-T Partition Value Only | deltaR | 4.20 ± 0.03 | 2688 | 64.1 ± 0.0 | 158414 | 17.05 ± 0.20 |
| SAL-T No Conv Partition Key Only | pt | 2.88 ± 0.03 | 2649 | 55.5 ± 0.0 | 118478 | 16.14 ± 0.16 |
| SAL-T No Conv Partition Key Only | deltaR | 2.92 ± 0.09 | 2649 | 55.5 ± 0.0 | 118478 | 16.42 ± 0.16 |
| SAL-T Partition Key Only | deltaR | 3.66 ± 0.43 | 2688 | 64.1 ± 0.0 | 158414 | 16.75 ± 0.02 |
| Linformer | pt | 2.91 ± 0.02 | 3033 | 54.1 ± 0.0 | 118478 | 16.26 ± 0.21 |
| Linformer | deltaR | 2.94 ± 0.05 | 3033 | 54.1 ± 0.0 | 118478 | 16.24 ± 0.43 |
| SAL-T No Partition | deltaR | 3.60 ± 0.20 | 3072 | 63.1 ± 0.0 | 158414 | 16.90 ± 0.26 |
| SAL-T No Partition | pt | 3.63 ± 0.21 | 3072 | 63.1 ± 0.0 | 158414 | 17.11 ± 0.39 |
| SAL-T Partition Key Only | pt | 3.74 ± 0.42 | 2688 | 64.1 ± 0.0 | 158414 | 17.39 ± 0.43 |
| Transformer | deltaR | 2.21 ± 0.38 | 2009 | 221.5 ± 0.0 | 197326 | 16.08 ± 0.15 |
| SAL-T No Conv Partition Value Only | kt | 3.02 ± 0.44 | 2649 | 55.5 ± 0.0 | 118478 | 16.51 ± 0.17 |
| SAL-T No Conv | kt | 3.34 ± 0.17 | 2265 | 53.0 ± 0.0 | 118478 | 16.20 ± 0.25 |
| SAL-T No Conv Partition Key Only | kt | 2.92 ± 0.03 | 2649 | 55.5 ± 0.0 | 118478 | 16.26 ± 0.16 |
| SAL-T Share EF | kt | 3.89 ± 0.31 | 2176 | 65.5 ± 0.0 | 158414 | 17.38 ± 0.25 |
| SAL-T Partition Key Only | kt | 3.69 ± 0.46 | 2688 | 64.1 ± 0.0 | 158414 | 17.21 ± 0.11 |
| SAL-T Partition Value Only | kt | 4.18 ± 0.05 | 2688 | 64.1 ± 0.0 | 158414 | 16.92 ± 0.14 |
| SAL-T | kt | 4.52 ± 0.32 | 2304 | 65.5 ± 0.0 | 158414 | 17.22 ± 0.25 |
| SAL-T No Partition | kt | 3.61 ± 0.19 | 3072 | 66.43 ± 5.77 | 158414 | 17.32 ± 0.12 |
| Transformer | kt | 2.08 ± 0.35 | 2009 | 221.5 ± 0.0 | 197326 | 15.96 ± 0.22 |
| Transformer | pt | 2.25 ± 0.37 | 2009 | 221.5 ± 0.0 | 197326 | 16.25 ± 0.14 |
| Linformer | kt | 2.93 ± 0.08 | 3033 | 54.1 ± 0.0 | 118478 | 16.38 ± 0.07 |

Table 14: Performance metrics for 150-particle HLS4ML dataset

| Model | Sort | Test Accuracy [%] | ROC AUC | Avg 1/FPR |
|---|---|---|---|---|
| SAL-T Share EF | pt | 73.86 ± 0.00 | 0.93 ± 0.01 | 13.52 ± 3.17 |
| SAL-T Share EF | deltaR | 74.16 ± 0.00 | 0.93 ± 0.01 | 13.51 ± 4.14 |
| SAL-T No Conv | deltaR | 76.08 ± 0.00 | 0.94 ± 0.00 | 16.05 ± 0.70 |
| SAL-T No Conv | pt | 76.91 ± 0.00 | 0.94 ± 0.00 | 17.90 ± 2.64 |
| SAL-T Partition Value Only | deltaR | 77.08 ± 0.00 | 0.94 ± 0.00 | 18.27 ± 2.73 |
| SAL-T No Conv Partition Value Only | pt | 77.61 ± 0.00 | 0.95 ± 0.00 | 19.53 ± 3.26 |
| SAL-T Partition Value Only | pt | 77.72 ± 0.00 | 0.95 ± 0.00 | 20.46 ± 3.47 |
| SAL-T No Conv Partition Value Only | deltaR | 78.15 ± 0.00 | 0.95 ± 0.00 | 20.91 ± 1.22 |
| SAL-T | deltaR | 78.25 ± 0.00 | 0.95 ± 0.00 | 20.64 ± 1.74 |
| SAL-T | pt | 78.82 ± 0.00 | 0.95 ± 0.00 | 23.45 ± 2.12 |
| SAL-T Partition Key Only | pt | 79.55 ± 0.00 | 0.95 ± 0.00 | 26.48 ± 1.90 |
| SAL-T No Partition | deltaR | 79.72 ± 0.00 | 0.95 ± 0.00 | 27.40 ± 0.74 |
| Linformer | deltaR | 79.78 ± 0.00 | 0.95 ± 0.00 | 27.66 ± 0.84 |
| SAL-T No Conv Partition Key Only | pt | 79.79 ± 0.00 | 0.95 ± 0.00 | 27.78 ± 1.54 |
| SAL-T No Partition | pt | 79.80 ± 0.00 | 0.95 ± 0.00 | 27.68 ± 0.67 |
| SAL-T No Conv Partition Key Only | deltaR | 79.81 ± 0.00 | 0.95 ± 0.00 | 27.81 ± 1.53 |
| SAL-T Partition Key Only | deltaR | 79.83 ± 0.00 | 0.95 ± 0.00 | 27.58 ± 2.69 |
| Linformer | pt | 79.90 ± 0.00 | 0.95 ± 0.00 | 28.06 ± 0.58 |
| Linformer | kt | 81.00 ± 0.00 | 0.96 ± 0.00 | 38.41 ± 0.54 |
| SAL-T No Partition | kt | 81.02 ± 0.00 | 0.96 ± 0.00 | 39.99 ± 0.65 |
| SAL-T Partition Key Only | kt | 81.03 ± 0.00 | 0.96 ± 0.00 | 38.61 ± 1.19 |
| SAL-T Partition Value Only | kt | 81.04 ± 0.00 | 0.96 ± 0.00 | 39.17 ± 0.76 |
| SAL-T Partition Key Only | kt | 81.05 ± 0.00 | 0.96 ± 0.00 | 39.04 ± 1.43 |
| SAL-T No Conv | kt | 81.09 ± 0.00 | 0.96 ± 0.00 | 39.32 ± 1.06 |
| SAL-T Share EF | kt | 81.12 ± 0.00 | 0.96 ± 0.00 | 39.78 ± 0.53 |
| SAL-T No Conv Partition Value Only | kt | 81.16 ± 0.00 | 0.96 ± 0.00 | 39.72 ± 0.48 |
| SAL-T | kt | 81.18 ± 0.00 | 0.96 ± 0.00 | 40.78 ± 0.57 |
| Transformer | pt | 81.27 ± 0.00 | 0.96 ± 0.00 | 42.02 ± 0.71 |

Table 15: Efficiency metrics for 150-particle HLS4ML dataset

| Model | Sort | Training Time [h] | Params | GPU Peak [MB] | FLOPs | Inference Time [μs] |
|---|---|---|---|---|---|---|
| SAL-T Share EF | pt | 7.40 ± 0.37 | 2656 | 303.4 ± 0.0 | 739918 | 27.91 ± 0.32 |
| SAL-T Share EF | deltaR | 7.33 ± 0.31 | 2656 | 303.4 ± 0.0 | 739918 | 27.63 ± 0.40 |
| SAL-T No Conv | deltaR | 5.07 ± 0.32 | 3225 | 226.2 ± 0.0 | 552718 | 22.83 ± 0.84 |
| SAL-T No Conv | pt | 5.19 ± 0.21 | 3225 | 226.2 ± 0.0 | 552718 | 22.71 ± 0.17 |
| SAL-T Partition Value Only | deltaR | 7.40 ± 0.09 | 5056 | 296.4 ± 0.0 | 739918 | 27.70 ± 0.58 |
| SAL-T No Conv Partition Value Only | pt | 4.74 ± 0.20 | 5017 | 252.9 ± 0.0 | 552718 | 23.01 ± 0.40 |
| SAL-T Partition Value Only | pt | 7.49 ± 0.12 | 5056 | 296.4 ± 0.0 | 739918 | 27.29 ± 0.47 |
| SAL-T No Conv Partition Value Only | deltaR | 4.68 ± 0.21 | 5017 | 252.9 ± 0.0 | 552718 | 22.24 ± 0.45 |
| SAL-T | deltaR | 7.63 ± 0.41 | 3264 | 303.4 ± 0.0 | 739918 | 27.69 ± 0.32 |
| SAL-T | pt | 7.84 ± 0.29 | 3264 | 303.4 ± 0.0 | 739918 | 27.39 ± 0.39 |
| SAL-T Partition Key Only | pt | 7.04 ± 0.37 | 5056 | 296.4 ± 0.0 | 739918 | 27.04 ± 0.24 |
| SAL-T No Partition | deltaR | 7.02 ± 0.20 | 6848 | 296.4 ± 0.0 | 739918 | 27.06 ± 0.17 |
| Linformer | deltaR | 4.66 ± 0.05 | 6809 | 245.8 ± 0.0 | 552718 | 22.38 ± 0.33 |
| SAL-T No Conv Partition Key Only | pt | 4.70 ± 0.01 | 5017 | 253.4 ± 0.0 | 552718 | 22.64 ± 0.39 |
| SAL-T No Partition | pt | 6.99 ± 0.19 | 6848 | 296.4 ± 0.0 | 739918 | 27.15 ± 0.33 |
| SAL-T No Conv Partition Key Only | deltaR | 4.68 ± 0.06 | 5017 | 253.4 ± 0.0 | 552718 | 22.02 ± 0.19 |
| SAL-T Partition Key Only | deltaR | 7.03 ± 0.37 | 5056 | 296.4 ± 0.0 | 739918 | 27.08 ± 0.54 |
| Linformer | pt | 4.57 ± 0.04 | 6809 | 245.8 ± 0.0 | 552718 | 22.54 ± 0.42 |
| Linformer | kt | 4.64 ± 0.01 | 6809 | 245.8 ± 0.0 | 552718 | 22.44 ± 0.30 |
| SAL-T No Partition | kt | 7.00 ± 0.22 | 6848 | 296.4 ± 0.0 | 739918 | 27.06 ± 0.33 |
| SAL-T Partition Key Only | kt | 4.64 ± 0.02 | 5017 | 253.4 ± 0.0 | 552718 | 22.73 ± 0.47 |
| SAL-T Partition Value Only | kt | 7.48 ± 0.15 | 5056 | 296.4 ± 0.0 | 739918 | 27.58 ± 0.44 |
| SAL-T Partition Key Only | kt | 7.03 ± 0.36 | 5056 | 296.4 ± 0.0 | 739918 | 27.23 ± 0.12 |
| SAL-T No Conv | kt | 5.00 ± 0.14 | 3225 | 226.2 ± 0.0 | 552718 | 22.73 ± 0.34 |
| SAL-T Share EF | kt | 7.27 ± 0.38 | 2656 | 303.4 ± 0.0 | 739918 | 27.79 ± 0.45 |
| SAL-T No Conv Partition Value Only | kt | 4.75 ± 0.25 | 5017 | 252.9 ± 0.0 | 552718 | 22.30 ± 0.37 |
| SAL-T | kt | 7.66 ± 0.36 | 3264 | 303.4 ± 0.0 | 739918 | 27.51 ± 0.35 |
| Transformer | pt | 1.49 ± 0.12 | 2009 | 4357.1 ± 0.0 | 2479918 | 31.03 ± 0.29 |

Table 16: Model performance comparison on the Quark Gluon dataset including training time. Best metrics between SAL-T and Linformer are highlighted in bold. In the case where two metrics are within statistical uncertainty, neither is highlighted. All models are sorted by $k_{\mathrm{T}}$.

| Layers | Model | # Params | FLOPs | Training Time [h] | Accuracy [%] | ROC AUC | 1/FPR@0.8TPR |
|---|---|---|---|---|---|---|---|
| | SAL-T | 3196 | 739761 | 6.19 ± 1.00 | 81.34 ± 0.05 | 0.8888 ± 0.0005 | 5.7570 ± 0.0475 |
| 1 | Transformer | 1941 | 2479761 | 7.23 ± 2.68 | 81.64 ± 0.03 | 0.8913 ± 0.0001 | 5.9820 ± 0.0269 |
| | vanilla | 6741 | 552561 | 3.67 ± 0.74 | 81.36 ± 0.01 | 0.8882 ± 0.0001 | 5.7713 ± 0.0110 |
| | SAL-T | 6171 | 2095761 | 9.81 ± 1.45 | 81.77 ± 0.00 | 0.8925 ± 0.0005 | 6.0725 ± 0.0106 |
| 2 | Transformer | 3529 | 4942161 | 11.51 ± 3.77 | 82.08 ± 0.02 | 0.8956 ± 0.0003 | 6.318 ± 0.0155 |
| | vanilla | 13129 | 1087761 | 7.20 ± 2.37 | 81.60 ± 0.09 | 0.8905 ± 0.0007 | 5.9410 ± 0.0529 |

