# OpenReview forum: "Spatially Aware Linear Transformer (SAL-T) for Efficient Particle Jet Identification"
_ICLR.cc/2026/Conference — Submitted to ICLR 2026_

### Official Review · Reviewer_shVZ · 2025-10-27

**Soundness:** 2
**Presentation:** 2
**Contribution:** 2
**Rating:** 2
**Confidence:** 5

**Summary:**

The paper tackles particle jet tagging. The proposed method boils down to
1) sorting the set of points given problem-specific metric;
2) splitting the sorted sequence into chunks and computing a representation of the chunk;
3) computing attention from each token to the chunks;
4) applying depthwise convolutions on attention logits to capture local correlations.

The method is evaluated on jet tagging datasets (hls4ml, Top Tagging, Quark-Gluon) and ModelNet10, demonstrating improved accuracy over Linformer with similar computational efficiency. The experimental evaluation has significant methodological flaws that prevent drawing reliable conclusions: (1) unfair memory comparisons without Flash Attention, (2) uncontrolled parameter counts confounding performance, (3) missing comparisons to relevant SOTA point cloud methods.

**Strengths:**

- The approach is well motivated - high amount of data and the scale of it make using a standard transformer for on-fly predictions unrealistic.
- The paper addresses a genuine practical need for real-time jet tagging at the HL-LHC, and the focus on trigger-deployable models (microsecond latency, low memory) is valuable for the HEP community.
- The metric used for data organization is theoretically-motivated and taken from jet clustering algorithms.
- Spatial partitioning aligns with the physical intuition that nearby particles in (Δη, Δφ) space are related.
- Ablation studies are helpful, specifically table 1 shows that k_T sampling achieves better performance than p_t sorting.
- The approach outperforms the main baseline - Linformer - including when the depth is increased.

**Weaknesses:**

- My main concern is the limited choice of baselines for comparison:
  - The approach feels spiritually close to Transolver [1], specifically, "computing attention between regions of physical significance". Yet, the comparison is missing from the baselines and is not even mentioned in related works.
  - The approach is close to Erwin [2], which sorts particles via ball trees and computes attention over groups (balls). The comparison is yet again missing.
  - Comparison to SOTA linear point cloud transformers such as Erwin [2] and PTv3 [3] is missing, even though both have linear complexity and are designed to handle point cloud data.
  - If I understand correctly, each partition is projected to a single vector, and then attention is computed from points to projections. That is very close to what UPT does [4].
- The approach uses simple sorting to organize data. While it seems to work empirically for the dataset authors use, I am not confident that the resulting partitions always make sense. For example, assume we have a N particles, and 3/4 particles are in one half of the domain, 1/4 particles in another. Using sorting + partitioning (proposed method) would generate one proper partition and one disjoint partition where locality assumption is violated. PTv3 handles the case by adaptive voxelization and Erwin uses ball trees with dummy nodes to specifically handle such cases. I believe some statistical analysis of partitioning across any dataset would strengthen confidence in this design choice.
- The paper does not seem to use flash attention, which results in high Peak GPU memory. For this reason, I cannot accept the value and conclusion draw from it - flash attention is linear in memory and would not achieve such high values.
- The strategy of choosing bold numbers is strange to me. Transformer is consistently better than either of methods, so I do believe it should be highlighted.
- Table 7 shows that accuracy plateaus at p=4 partitions and slightly decreases beyond that (p=8: 81.10%, p=16: 80.97% vs p=4: 81.18%). This suggests the method does not scale with increased compute budget, which is concerning. The authors should discuss why additional partitions hurt performance - is it overfitting, optimization difficulties, or a fundamental limitation of the partitioning strategy?
- Regarding Table 3 - why is the number of params different for each model? I believe a more fair comparison would be to use the same number of parameters and report FLOPs and runtime.
- Additionally, in Table 3, the Transformer achieves 92.91% accuracy with only 2k parameters, while SAL-T achieves 92.52% with 3.5k parameters. This suggests the Transformer is actually more parameter-efficient than SAL-T, contradicting the paper's efficiency narrative.


[1] Wu, Haixu et al. “Transolver: A Fast Transformer Solver for PDEs on General Geometries.” ArXiv abs/2402.02366 (2024): n. pag.

[2] Zhdanov, Maksim et al. “Erwin: A Tree-based Hierarchical Transformer for Large-scale Physical Systems.” ArXiv abs/2502.17019 (2025): n. pag.

[3] Wu, Xiaoyang et al. “Point Transformer V3: Simpler, Faster, Stronger.” 2024 IEEE/CVF Conference on Computer Vision and Pattern Recognition (CVPR) (2023): 4840-4851.

[4] Alkin, Benedikt et al. “Universal Physics Transformers.” ArXiv abs/2402.12365 (2024): n. pag.

**Questions:**

- How is the proposed approach different from Erwin [2], which also sort particles into partitions and computes attention over those partitions? Originally, I believe it uses euclidean metric, but it might be possible to change it to an arbitrary metric, including the one used by authors.
- Would it be possible to expand the set of baselines to [1,2,3]?
- Do you use flash-attn for transformer implementation? If not, that would explain high peak memory. Therefore, I would ask for reporting values for transformers with flash-attn, since it is a standard approach nowadays, not naive scaled dot product.

---

> ### Author Response · Authors · 2025-11-18
> **Response to Reviewer shVZ (part 1)**
>
> Thank you for the detailed feedback. Additional context is important regarding the intended deployment environment for this work. Below we provide clarification and additional context relevant to the CMS Level-1 Trigger (L1T) for the High-Luminosity LHC (HL-LHC), which is the target deployment environment for this work.
>
> At the HL-LHC, the beam crossing rate will be 40 MHz, and the L1T must reduce throughput by roughly a factor of 400 within an end-to-end latency budget on the order of microseconds. These constraints permit only extremely compact neural network architectures that can be efficiently synthesized on FPGA or AI-engine hardware.
>
> The CMS detector alone produces on the order of 1 PB of raw data (roughly a billion collision events) every second , of which only a tiny fraction can be stored. The L1T must make real-time decisions about which events to keep for physics analysis. Because of the scale of our data, even a fractional improvement in trigger performance has a substantial scientific payoff. A gain of a few tenths of a percent in classification accuracy corresponds to recovering **millions** of rare and interesting collision events that would otherwise be lost, directly increasing the discovery potential of the experiment. Tiny-ML for L1T is a very young and largely unexplored field, and the strongest deployed models in LHC trigger production today remain dense feed-forward networks and small CNNs.
>
> ## Relation to Erwin, Transolver, UPT, and PTv3
>
> While there are some structural similarities to hierarchical or region-aware point-cloud transformers, the computational assumptions and scientific objectives are fundamentally different.
>
> ### Erwin
> Erwin relies on hierarchical grouping using ball-tree clustering. Any clustering-based approach is far too slow for the Level-1 Trigger. If clustering were possible in this latency envelope, particle physics already has highly optimized clustering algorithms ($k_T$, anti-$k_T$, Cambridge/Aachen), but even those are too slow for the L1T. SAL-T instead uses sorting by the $k_T$ scalar, which provides a fast physics-motivated ordering without needing explicit clustering.
>
> ### UPT
> UPT is built around neural operators, which are fundamentally different from standard neural networks. Neural operators are a completely different paradigm of information processing, and is not a good comparison when optimizing neural networks.
>
> ### Transolver
> Transolver focuses on PDE-based simulation, not jet tagging. Good performance on PDE solving has no correlation to particle-jet classification. The scientific objective and numerical behavior are entirely different.
>
> ### Point Transformer V3
> We are producing benchmarks on Pointnet and Point transformer, and will post them in the rebuttals in the coming weeks.
>
> ---
>
> ## Baselines and expanded comparisons
>
> The baselines used in the paper are those that are actually deployable in the Level-1 Trigger today. Comparing to point-cloud models with 800k–3M parameters is not meaningful without reducing them by several orders of magnitude and synthesizing under identical hardware constraints. We are currently benchmarking compressed PointNet and Point Transformer V3 models (reduced to ~3k parameters, which the same architecture, to match SAL-T). These additional results will be included in the next revision in the coming weeks.
>
> As a point of reference, we also compare against a tiny 1-layer Particle Transformer[1] whcih is one of the strongest jet-tagging architectures being used in the HEP community—under identical constraints. The only model that outperforms ParT (LGATR) is ten times slower than ParT in our parameter count range. Even aggressively downsized, it remains too expensive for L1T use:
>
> | Model         | FLOPs     | Inference Time [μs]  | Peak GPU Mem [MB] | Accuracy [%] | ROC AUC | 1/FPR@0.8TPR |
> |---------------|-----------|----------------------|-------------------|--------------|---------|---------------|
> | 1-layer-ParT  | 8,217,868 | **211.20 ± 0.84**        | **OOM**               | 81.42 ± 0.19 | 0.9599  | 43.16 ± 1.67  |
> | SAL-T     | 739,918   | 26.78 ± 0.13   | 303.4         | 81.18 ± 0.03 | 0.9593 | 40.78 ± 0.57 |
> | transformer   | 2,479,918 | 29.61 ± 0.12         | 4,357.1           | 81.27 ± 0.08 | 0.9589  | 42.02 ± 0.71  |
>
> Even with a single layer, ParT is more than 10× more expensive in FLOPs and nearly an order of magnitude slower in inference, demonstrating the necessity of a specifically engineered architecture.

---

> ### Author Response · Authors · 2025-11-18
> **Response to Reviewer shVZ (part 2)**
>
> ## Partition strategy and scaling behavior
>
> The decline when increasing partitions beyond p=4 is expected since further partitioning spreads information too thin relative to the small projection dimension available. For instance, with 8 partitions, there would be ~19 particle per partition, while with 16 there would be around 9. We found that SAL-T is best performant with a 4 partitions. Number of partitions is a parameter that must be adapted to the dataset at hand. Some application cases may appreciate higher numbers of partitions, but in our case 4 is the ideal.
>
> In addition, to address the last partition having very few particles. We conducted two additonal experiments that increase the number of particles that partition 4 sees. First, we put an equal number of particles in each partition, and pad the rest of the partition with zero. We find that it performs worse than our current method.
>
> | Model                | Accuracy [%] | ROC AUC | 1/FPR@0.8TPR |
> |----------------------|--------------|---------|--------------|
> | **SAL-T**            | **81.18 ± 0.03** | **0.9593** | **40.78 ± 0.57** |
> | Equal Partition| 80.87 ± 0.21 | 0.9582 | 36.82 ± 2.51 |
>
> In addition, we also shuffle partitions during training so that each partition sees a variety of particles from different $k_T$ bins. Partition shuffling experiments  also demonstrate that the default partitioning consistently performs best overall:
>
> | Model | Overall [%] | Bin 1 [%] | Bin 2 [%] | Bin 3 [%] | Bin 4 [%] |
> |--------|------------|-----------|-----------|-----------|-----------|
> | shuffle all partitions | 80.38 ± 0.08 | 80.98 ± 0.09 | 79.66 ± 0.08 | 81.76 ± 0.47 | **95.12 ± 0.00** |
> | shuffle partitions 2, 3, 4 | 80.94 ± 0.11 | 81.41 ± 0.13 | 80.40 ± 0.09 | **82.08 ± 0.46** | **95.12 ± 0.00** |
> | shuffle partitions 3 and 4 | 81.11 ± 0.14 | 81.58 ± 0.09 | 80.51 ± 0.18 | 81.99 ± 0.60 | 89.8 ± 8.96 |
> | **default (not shuffled)** | **81.18 ± 0.03** | **81.67 ± 0.01** | **80.60 ± 0.06** | 81.60 ± 0.20 | 89.43 ± 6.14 |
>
> ## Parameter count differences and efficiency claims
>
> The small increase in parameter count in SAL-T (compared to the baseline Transformer) comes **only** from the additional linear projection layers used to compress the original sequence length into a smaller dimension. These projection matrices replace part of the quadratic attention computation with a linear approximation, and they are the sole source of parameter differences. The core architecture — number of layers, hidden layer size, attention heads, and feed-forward components — remains identical across Transformer, Linformer, and SAL-T.
>
> Because these added parameters are specifically used to **compress the sequence and reduce attention complexity**, they actually improve computational efficiency. In this setting, parameter count does not reflect computational cost: the projection layers introduce a small number of extra weights, but they drastically reduce FLOPs, memory access, and inference latency. Thus, raw parameter count is not a meaningful indicator of efficiency for models like Linformer and SAL-T, where the additional parameters directly enable the linear-attention approximation.
>
> The baseline Transformer appears more “parameter efficient” only because it performs full quadratic attention on the uncompressed sequence, an operation that is computationally far more expensive. If the Transformer were reduced to the same order of magnitude in compuatational resource consumption as Linformer or SAL-T, it would require a drastically smaller hidden dimension, leading to a substantial loss in accuracy. In contrast, Linformer and SAL-T use their extra projection parameters to lower the cost of attention while maintaining expressive capacity.
>
> Finally, for deployment on custom hardware such as FPGAs or AI-engine accelerators, **FLOPs, not parameter count, are the primary bottleneck**. Sequence compression directly targets FLOPs, which is why the modest parameter increases in Linformer and SAL-T translate into disproportionately large gains in hardware efficiency.
>
> For the upcoming benchmakrs with PointNet and Point Transformer, we will be using the same philosophy as above where we scale the models down, while using the same architecture.
>
>
> ## On Flash Attention and memory numbers
>
> Flash Attention is a hardware-specific optimization designed for GPUS, which is not relevant in the context of deployment on FPGA, since our implementations are built through hls4ml for FPGA inference, not GPU kernels. We are benchmarking Flash Attention on 1080-Ti GPU for completeness and will include them in the upcoming week in the rebuttal.
>
> ---
>
> ## Closing
>
> Thank you again for the constructive feedback. We will include the following in the upcoming weeks in the rebuttal:
> - compressed PointNet and Point Transformer V3 comparisons
> - Flash-Attention baseline on GPU
> - updated ModelNet10 results
>
> [1] Qu et al, Particle Transformer for Jet Tagging, ICML 2022

---

> > ### Comment · Reviewer_shVZ · 2025-11-21
> > **Rebuttal acknowledgement by Reviewer shVZ**
> >
> > Dear authors, thank you for taking the time to address my questions. I have carefully read your response and have a few follow-up questions that would help clarify things for me.
> >
> > 1. I would respectfully disagree with your dismissal of neural operators (UPT) and Transolver. They are also universal approximators, just like the Linformer on which you based your work. Moreover, they have sub-linear complexity, which is relevant at the scale at which you aim to operate. Hence, I would suggest testing them. Transolver has a clean official implementation and should not take too much time to benchmark. UPT is similar in spirit -- using attention to regions -- so why would it be inapplicable? Do you think it would be possible to provide those benchmarks?
> >
> > 2. I would also disagree that FlashAttention is hardware-specific in the way you describe. It is essentially an algorithm that assumes hierarchical memory on your device. I am not very familiar with FPGAs, but I assume they have hierarchical memory? If so, FlashAttention is definitely viable in principle. Just to clarify - I am absolutely not asking you to implement FA on FPGA, but it is something to keep in mind: standard attention is only quadratic in compute, not in memory, when properly implemented.
> >
> > 3. Furthermore, in Table 3, you report peak GPU memory. Therefore, I see no obstacle to applying FlashAttention in your current setup -- doing so would only be fair, in my opinion, and would indicate whether there should be an effort in the community to implement FPGA support for it. Right now, it appears that you are indicating high memory consumption as a key limitation of Transformers, when in fact it is a limitation of the specific implementation you chose, not an inherent limitation of the Transformer architecture itself. Would you agree?
> >
> > I will therefore retain my score for the time being until the promised benchmarks (PointNet, PTv3, and FlashAttention) are provided.

---

> ### Author Response · Authors · 2025-11-26
> **Response to reviewer shVZ****
>
> Thank you for requesting broader baselines and clarification regarding FlashAttention. In addition to the PointTransformer and PointNet results included above, we also evaluated a compact Transolver variant adapted to the same latency-constrained regime (single block, reduced width, identical training schedule). Transolver when under trigger-scale constraints it is slower than SAL-T and Transformer while achieving noticeably lower background rejection.
>
> **Transolver comparison (hls4ml jets)**
>
> | Model | Inference Time (µs) | Peak GPU Mem (MB) | Accuracy (%) | 1/FPR @ 0.8 TPR |
> |--------|-----------------------:|----------------------:|----------------:|-----------------------:|
> | Transolver | 88.44 ± 0.33 | 2,771.4 | 80.13 ± 0.09 | 37.05 ± 0.52 |
> | **Transformer** | **29.61 ± 0.12** | **4,357.1** | **81.27 ± 0.08** | **42.02 ± 0.71** |
> | **SAL-T** | **26.78 ± 0.13** | **303.4** | **81.18 ± 0.03** | **40.78 ± 0.57** |
>
> Transolver sits between PointTransformer-small and PointNet in both performance and latency but does not approach Transformer or SAL-T. It cannot be scaled up further without exceeding realistic resource footprints.
>
> ---
>
> ### FlashAttention
>
> We additionally benchmarked FlashAttention for the compressed Transformer baseline using PyTorch. The main paper uses TensorFlow because it integrates with the CMS L1T + hls4ml deployment pipeline, and FlashAttention is not available there. The paper reports standard attention results since they reflect the realistic implementation path for FPGA-oriented deployment, which is done in tensorflow keras, which does not support flashattention yet. The FlashAttention results are included below strictly for comparison.
>
> **PyTorch 1-Layer Attention Benchmarks (A10 GPU)**
>
> | Model | Attention Mode | Inference Time (µs/seq) | Peak GPU Mem (MB) |
> |--------|----------------|-------------------------:|-------------------:|
> | Transformer-1L | Flash-Attention | **4.42** | 52.3 |
> | Transformer-1L | standard | **7.56** | 389.2 |
> | **SAL-T-1L** | standard | **3.99** | **62.2** |
>
> ---
>
> ### FPGA considerations
>
> While FlashAttention-like tiling and streaming approaches *can* be implemented on FPGA, their practicality depends heavily on chip architecture and routing. Existing demonstrations (e.g., tiled attention acceleration work from the University of Washington & Taiwan collaboration) show feasibility, but they rely on multi-stage accumulation and loop unrolling rather than single-pass matrix multiplication. This reduces peak memory but can increase latency, since the attention computation must be staged across cycles. Under 40 MHz Level-1 trigger constraints, latency (balanced with background rejection and accuracy) is the limiting resource. By design, SAL-T avoids the quadratic attention matrix entirely, which removes the need for tiling logic or multi-phase accumulation and simplifies hardware scheduling. The reduced memory footprint decreases reliance on large BRAM/URAM buffers which introduce additional pipeline staging and synchronization overhead, which can increase latency.
>
> In LHC (and other particle accelerators), latency is one of the most important benchmarks because the accelerator must be able to process the incoming collision data, filter then store the data before the next batch of particles collide. If the model is unable to do it within this latency (40MhZ), lots of valuable particle physics data may be thrown away. SAL-T provides a balance of latency, accuracy, and background rejection specifically for this important task.
>
>
> --
>
> **Reference**
>
> Hauck et al., *Transformer Acceleration on FPGA*, University of Washington & Taiwan collaboration.
> https://people.ece.uw.edu/hauck/publications/Transformer_Taiwan.pdf

---

> > ### Comment · Reviewer_shVZ · 2025-11-27
> > **Response by Reviewer shVZ**
> >
> > I am grateful to the authors for their effort and comprehensive response. The additional comparisons significantly strengthen the paper. That being said, I believe the submission would benefit from substantial revision before it meets the bar for a top-tier ML conference.
> >
> > **Primary Concern: Inconsistent Experimental Framework**
> >
> > Since the main contribution is an architectural improvement, comparisons to baselines should be conducted on a fair and consistent basis. Currently, the comparisons lack a unified framework - there is no single controlled variable (FLOPs, inference time, or peak memory) held constant across all baselines, making it difficult to draw definitive conclusions about the relative merits of each approach. I would suggest restructuring the experimental evaluation around controlled comparisons:
> > - *Fixed compute budget*: Given a specific FLOPs budget, how does each model perform in terms of accuracy, memory footprint, and runtime?
> > - *Fixed latency constraint*: Since runtime is the primary deployment concern, given the same forward pass latency, what accuracy does each model achieve and at what memory cost?
> >
> > In other words, fix one property (FLOPs, latency, or memory) and systematically vary the others. This would provide a clearer picture of the trade-offs and make the advantages of SAL-T more transparent and convincing.
> >
> > Additionally, the FlashAttention results you provided demonstrate that memory efficiency is implementation-dependent rather than an inherent architectural limitation of Transformers. While I understand FlashAttention is not currently available in your TensorFlow/hls4ml pipeline, this distinction should be explicitly stated in the paper to avoid misleading readers about the fundamental capabilities of different architectures.
> >
> > **Recommendation:**
> >
> > I believe the authors have all the necessary components for a high-quality submission: the problem is important, the constraints are well-motivated, and you've now gathered comprehensive baseline comparisons. However, the work would benefit from substantial revision -- particularly a redesigned experimental framework with controlled comparisons -- before it is ready for publication at a top-tier venue.
> >
> > I will raise my score to weak reject (4) to acknowledge the effort but would encourage the authors to take the time to refine this work thoughtfully and submit the revised version to a future conference. With these improvements, I believe this could become a strong contribution to both the ML and HEP communities.

---

> ### Author Response · Authors · 2025-11-27
> **Response to reviewer shvz**
>
> Thank you for the thoughtful feedback and for highlighting the importance of a clearly structured comparison framework. We agree that presenting results under fixed compute or latency budgets would strengthen the clarity of trade-offs and improve the manuscript. We appreciate this suggestion, and in the revision we will reorganize the results to explicitly present accuracy–latency and accuracy–FLOPs Pareto plots, making the relationships between performance and resource constraints more transparent.
>
> We would also like to clarify that our existing evaluation methodology follows the standard practice established in the efficient-Transformer literature, such as Reformer, Linformer, Performer, and Longformer. In these works, the experimental protocol keeps the core model architecture and training configuration constant while varying only the attention mechanism, and reports accuracy, latency, FLOPs, and memory as empirical outcomes rather than fixed control variables. This comparison strategy is standard because different attention mechanisms exhibit fundamentally different scaling behaviors—for example, $\(O(n^2)\)$ vs. $\(O(nk)\)$—making it infeasible to simultaneously equalize FLOPs, latency, and memory without altering model width or depth, which would confound representational capacity.
>
> Consistent with this methodology, our experiments keep the following hyper parameters constant:
> number of layers,hidden dimension and feed-forward dimension,number of attention heads,sequence length,input representation and dataset preprocessing,optimizer, batch size, and training schedule,hardware environment for latency and memory measurements.
> The only architectural difference among SAL-T, Linformer, and Transformer is the formulation of the attention mechanism itself. This comparison structure directly parallels the framework used in prior efficient-attention work. We also note that the same evaluation methodology was applied to the additional baselines introduced in the rebuttal, including Point Transformer, which we found to underperform compared to SALT without the physics informed modifications. Using this evaluation strategy, the data is able to ask that given the same architecture, what impact does this upgraded layer have on the FLOPS and latency? For SAL-T we show that it improves accuracy over Linformer, while reducing FLOPs and latency over transformer.
>
> That said, we agree with the reviewer that reorganizing the results to emphasize controlled trade-offs and Pareto efficiency would improve clarity, and we will adopt this recommendation in the revised manuscript while maintaining the standard comparison framework used in the efficient-Transformer literature.
>
> ---
>
> ### References
> [1] Kitaev, Nikita, Łukasz Kaiser, and Anselm Levskaya. "Reformer: The efficient transformer." arXiv preprint arXiv:2001.04451 (2020).
> [2] Wang, Sinong, et al. "Linformer: Self-attention with linear complexity." arXiv preprint arXiv:2006.04768 (2020)
> [3] Choromanski, Krzysztof, et al. "Rethinking attention with performers." arXiv preprint arXiv:2009.14794 (2020).
> [4] Beltagy, Iz, Matthew E. Peters, and Arman Cohan. "Longformer: The long-document transformer."

---

> > ### Author Response · Authors · 2025-12-03
> > **Compute Matching Benchmarks**
> >
> > To further address the reviewer’s request, we additionally conducted a new set of FLOPs-aligned benchmarks, where we adjusted the model configurations so that SAL-T and the Transformer operate under closely matched computational cost. These results are shown below:
> >
> > # Performance Table
> >
> > | Model               | Sort | Accuracy (%)        | Avg Background Rejection      |
> > |---------------------|------|----------------------|--------------------------------|
> > | **SALT**            | kt   | **81.18 ± 0.03**     | **40.78 ± 0.57**               |
> > | **SAL-T (No Conv)** | kt   | 81.09 ± 0.10         | 39.32 ± 1.06                   |
> > | **Transformer**     | pt (permutationally equivariant)   | 78.93 ± 0.51         | 26.19 ± 1.98                   |
> >
> >
> > # Model Size & Speed Table
> >
> > | Model              | # Params | FLOPs      | Inference Time (µs) |
> > |--------------------|----------|-----------------|----------------------|
> > | SAL-T               | 3,264    | 739,918                | 27.69 ± 0.32         |
> > | SAL-T (No Conv)    | 3,225    | 552,718                     | 22.73 ± 0.34         |
> > | Transformer        | 339      | 889,368               | 57.50 ± 0.17         |
> >  In the above results, we highlight SAL-T without the convolution layer (SAL-T no conv) when presenting compute matched benchmarks. This variant preserves the core physics informed, locality aware linear attention of SAL-T while using lower FLOPs, and fewer parameters, than the baselines.
> >
> > To ensure a fair comparison, we also include a scaled down Transformer whose FLOP count is matched to that of SAL-T. In order to do this, we needed to significantly reduce the size of the hidden layer of the native transformer. Even under this controlled setting, where the Transformer is explicitly reduced to operate within the same compute envelope, SAL-T no conv still achieves higher accuracy, stronger background rejection, and lower latency. In addition, SAL-T no conv has almost identical parameter counts, and inference time as the native Linformer. At this level, SAL-T no conv has better performance in accuracy and rejection, with lower flops thatn Linformer as well. Transformer is permutationally equivariant, so the sorting scheme does not impact performance.
> >
> > This shows that the performance improvements come from the SAL-T attention design itself rather than from the  additional computational budget, confirming that SAL-T provides a better accuracy and efficiency tradeoff than both Linformer and the Transformer under matched FLOPs.

---

### Official Review · Reviewer_Uj2d · 2025-10-31

**Soundness:** 2
**Presentation:** 2
**Contribution:** 1
**Rating:** 4
**Confidence:** 3

**Summary:**

The paper introduces a Spatially Aware Linear Transformer for jet tagging, with applicability to point-cloud classification. The method reduces redundant particle connectivities via a sort-and-partition mechanism, and applies a convolutional layer over the attention matrix to enhance locality by influencing nearby particles. Experiments report modest improvements over baseline models.

**Strengths:**

* The method outperforms baseline approaches.
* It reduces computational overhead relative to baselines while matching the accuracy of a standard Transformer.

**Weaknesses:**

1. Relation to TIE [1]. is also a Transformer-based method with improved attention mechanics.
2. Outdated baselines. The paper compares mainly to models before 2020. Please include stronger and more recent baselines from the same domain or with similar architectures (e.g., modern Transformer variants, locality-aware/self-attention sparsification methods, and recent point-cloud/jet-tagging Transformers) to contextualize performance.
3. Presentation and organization. The writing and layout can be improved for readability: Reduce excessive white space around tables/figures. Move Figure 2 closer to the method section where it is first referenced. Place Table 5 at the top of a page or immediately after its first mention.
4. Scalability and particle count. Current settings are relatively small (≈150 particles for jet tagging; ≈1k for point-cloud classification). Additional classification results, as suggested in [2], would further substantiate the method’s effectiveness.
5. Limited incremental gains and component effectiveness. Table 1 shows only modest improvements (e.g., +0.18% accuracy) and suggests that sorting contributes more than the convolution layer; in some settings (e.g., with the best sorting strategy $k_T$) in Table 9, removing convolution yields better accuracy. As a result, the convolution seems less effective and the novelty may reduce to only a sorting mechanism.

[1]. Shao, et al. Transformer with Implicit Edges for Particle-based Physics Simulation. ECCV 2022.
[2]. Zhang, et al. Deep Learning-based 3D Point Cloud Classification A Systematic Survey and Outlook. Displays 2023.

**Questions:**

Please refer to the weaknesses.

---

> ### Author Response · Authors · 2025-11-18
> **Response to Reviewer Uj2d (part 1)**
>
> Thank you for the detailed feedback. Several points raised require additional context related to the constraints and goals of the CMS Level-1 Trigger (L1T) system at the High-Luminosity LHC in addition to other up and coming trigger systems for particle accelerators, which is the intended deployment environment for this work.
>
> At the HL-LHC, CMS produces on the order of 1 PB of raw detector data (roughly 1 billion collision events) every second. Nearly all of it must be discarded in real time because only a few thousand events per second can be stored. The Level-1 Trigger must operate within a strict sub-microsecond latency budget and run on FPGAs or AI engines with extremely limited compute and memory. In this domain, even a few tenths of a percent improvement in jet classification accuracy corresponds to retaining **millions** of rare physics events. A small increase in accuracy would directly impact discovery potential, and greatly increase the number of viable candidates for analysis. Tiny-ML for trigger is a very young field, and the strongest deployed baselines in CMS detector production remain dense networks and simple convolutional architectures. More sophisticated offline models exist, but they cannot be used directly due to resource constraints. SAL-T is designed specifically for this real-time environment.
>
> ## Relation to TIE and recent works
>
> TIE focuses on implicit edge construction for particle-based simulation, where graph connectivity evolves dynamically between time steps. Although it is an interesting direction, the problem setting and constraints differ significantly from the L1T environment, where latency and fixed compute budget dominate. Our objective is not to capture long-range physics simulation dynamics, but to construct a model that fits within the FPGA resource envelope while improving the trigger performance. The spatial-aware linear attention mechanism introduced here addresses a fundamentally different task and need.
>
> In addition, our particle physics simulations are much different from the fluid simulations mentioned in the paper. Particle physics is completely dissimilar from fluid physics (although we both use point clouds), and we are not confident a similar method would work for large offline analysis models, much less for triggering. **Particle-based physics is not the same as particle physics.**
>
> ## Baselines and comparisons
>
> The baseline models chosen are those feasible for the Level-1 Trigger today. Models such as PointNet, Point Transformer, Point Transformer V2/V3, and similar point-cloud backbones operate with hundreds of thousands to millions of parameters. In contrast, our model contains roughly 3,000 parameters. Direct head-to-head comparison is not meaningful unless those models are reduced by orders of magnitude and synthesized under identical hardware constraints.
>
> We agree it is valuable to explore scaled-down versions of these architectures. We are currently benchmarking reduced PointNet and Point Transformer V3 on ModelNet10 and hls4ml-particle physics dataset configurations. These results will be included in the rebuttal in the comming weeks.
>
> We also performed a comparison with a tiny 1-layer Particle Transformer[1] (one of the strongest architectures in the jet-tagging domain), demonstrating that even aggressively downsizing such models leads to prohibitive compute and memory overhead for L1T:
>
> | Model         | FLOPs     | Inference Time [μs]  | Peak GPU Mem [MB] | Accuracy [%] | ROC AUC | 1/FPR@0.8TPR |
> |---------------|-----------|----------------------|-------------------|--------------|---------|---------------|
> | 1-layer-ParT  | 8,217,868 | **211.20 ± 0.84**        | **OOM**               | 81.42 ± 0.19 | 0.9599  | 43.16 ± 1.67  |
> | SAL-T     | 739,918   | 26.78 ± 0.13   | 303.4         | 81.18 ± 0.03 | 0.9593 | 40.78 ± 0.57 |
> | transformer   | 2,479,918 | 29.61 ± 0.12         | 4,357.1           | 81.27 ± 0.08 | 0.9589  | 42.02 ± 0.71  |
>
> Even at a single layer, ParT is more than 10 times more expensive than SAL-T in FLOPs.
>
> ## Convolution and component effectiveness
>
>  In Table 9, the case where convolution underperforms corresponds to a very low number of max particles in the sequence (16). 16 particles is used traditionally in older trigger systems, but in the coming HL-LHC, trigger algorithms will be able to handle more particles. At 150 particles (table 5), convolution improves classification performance.

---

> ### Author Response · Authors · 2025-11-18
> **Response to Reviewer Uj2d (part 2)**
>
> ## Scalability and particle count
>
> Particle multiplicities above 115 occur extremely rarely in typical proton–proton jets at 13–14 TeV. Typically, the number of particles is capped at 128, since these interactions happen so rarely. Thus a 1000 particle use case would not be very practical.
>
> Thank you for the constructive comments. We hope the additional context clarifies the motivation behind the design choices and the scope of our experiments. Updates with compressed PointNet and Point Transformer results will be added soon.
>
> [1] Qu et al, Particle Transformer for Jet Tagging, ICML 2022

---

### Official Review · Reviewer_x3Ew · 2025-11-01

**Soundness:** 3
**Presentation:** 3
**Contribution:** 2
**Rating:** 4
**Confidence:** 2

**Summary:**

The paper presents Spatially Aware Linear Transformer, an efficient model for high-energy physics tasks such as jet tagging, which utilizes a spatially aware linear attention mechanism. The model builds on the linformer architecture and enhances it with techniques inspired by jet physics, such as spatial sorting and convolutional attention. SAL-T effectively reduces the computational cost compared to traditional transformers while maintaining competitive performance on jet tagging benchmarks.

**Strengths:**

The integration of spatial sorting and convolutional attention is an innovative approach that addresses both performance and computational efficiency.

 SAL-T’s use of linear attention and partitioned projections significantly reduces complexity, making it viable for real-time systems like LHC triggers.

**Weaknesses:**

A significant limitation of this paper is its failure to compare SAL-T with other state-of-the-art models for 3D point cloud processing, such as PointNet [1], Point Transformer [2], and Point Transformer V2 [3]: Grouped Vector Attention and Partition-based Pooling. These models are well-established in the point cloud classification space and have demonstrated superior performance on benchmarks like ModelNet10. Since the jet tagging task essentially involves 3D point cloud inputs, the omission of these models for comparison raises concerns. A more thorough comparison with these models would have provided clearer insights into SAL-T's strengths and weaknesses within the broader context of point cloud processing.

Additionally, PointNet, Point Transformer, and Point Transformer V2 have been shown to outperform SAL-T on ModelNet10, which is a key benchmark for point cloud classification. Given that jet tagging inherently involves complex point cloud-like data, it is surprising that these models, which are specifically designed to process such data, were not included for comparison. This failure to test SAL-T against widely adopted point cloud backbones leaves a gap in the paper's evaluation, limiting the ability to fully assess SAL-T's effectiveness and competitiveness within the field of 3D point cloud processing.


[1] Pointnet: Deep learning on point sets for 3d classification and segmentation. ICCV 2017.
[2] Point transformer. ICCV 2021.
[3] Point transformer v2: Grouped vector attention and partition-based pooling. NeurIPS 2022.

**Questions:**

See weakness

---

> ### Author Response · Authors · 2025-11-18
> **Response to Reviewer x3Ew**
>
> Thank you for the constructive comments. We would like to provide clarification and additional context relevant to the CMS Level-1 Trigger (L1T) for the High-Luminosity LHC (HL-LHC), which is the target deployment environment for this work.
>
> At the HL-LHC, the beam crossing rate will be 40 MHz, and the L1T must reduce throughput by roughly a factor of 400 within an end-to-end latency budget on the order of microseconds. These constraints permit only extremely compact neural network architectures that can be efficiently synthesized on FPGA or AI-engine hardware.
>
> SAL-T is designed specifically for the trigger environments at the HL-LHC. Under these conditions, we are limited to extremely small models—on the order of only a few thousand parameters. SAL-T has roughly 3k parameters. By contrast, architectures such as PointNet, Point Transformer, and Point Transformer V2 are typically hundreds of thousands to millions of parameters (commonly around 800k–2M). Direct comparison in their standard form is not meaningful because they are far beyond what can be implemented at our model scale.
>
> We are working on Tiny-ML for particle physics trigger applications, which is a very young and largely unexplored field. At present, the only deployed baselines in CMS production are a dense network and a small CNN, and research is only beginning to explore transformer-style architectures under these extreme resource limits. Comparisons must reflect what can actually run under hardware constraints rather than models designed without them. Improvements in this area have enormous implications for particle discovery, since the CMS detector alone produces on the order of 1 PB of raw data (roughly a billion collision events) every second, of which only a tiny fraction can be stored. The L1T must make real-time decisions about which events to keep for physics analysis. Because of the scale of our data, even a fractional improvement in trigger performance has a substantial scientific payoff. A gain of a few tenths of a percent in classification accuracy corresponds to recovering **millions** of rare and interesting collision events that would otherwise be lost, directly increasing the discovery potential of the experiment.
>
> In addition, we benchmarked a tiny, scaled down one-layer Particle Transformer[1] model (one of the strongest jet-tagging architectures available) reduced as far as possible toward our target compute range. Even this aggressively reduced version remains far too heavy:
>
> | Model         | FLOPs     | Inference Time [μs]  | Peak GPU Mem [MB] | Accuracy [%] | ROC AUC | 1/FPR@0.8TPR |
> |---------------|-----------|----------------------|-------------------|--------------|---------|---------------|
> | 1-layer-ParT  | 8,217,868 | **211.20 ± 0.84**        | **OOM**               | 81.42 ± 0.19 | 0.9599  | 43.16 ± 1.67  |
> | SAL-T     | 739,918   | 26.78 ± 0.13   | 303.4         | 81.18 ± 0.03 | 0.9593 | 40.78 ± 0.57 |
> | transformer   | 2,479,918 | 29.61 ± 0.12         | 4,357.1           | 81.27 ± 0.08 | 0.9589  | 42.02 ± 0.71  |
>
> Although the small ParT variant achieves slightly higher accuracy, it is much more expensive than even the native transformer (inference being an order of magnitude slower and OOM on 1080-Ti GPU).
>
> Regarding point cloud baselines such as PointNet and Point Transformer, the published benchmarks on ModelNet10 use models that are orders of magnitude larger than SAL-T. It is not surprising that large models perform better when allowed hundreds of thousands to millions of parameters, and more importantly, many millions of FLOPS. The benchmarks on ModelNet10 are not intended to demonstrate state-of-the-art performance on generic point cloud classification, but rather to show that our proposed compression mechanism, combined with lightweight convolution, effectively takes advantage of spatial locality in the input.
>
> We are currently preparing additional results on hls4ml and ModelNet10 using aggressively reduced versions of PointNet and Point Transformer V3 scaled down to our parameter/FLOP range. These experiments are in progress and will be included in rebuttal in the coming weeks. We hope they will provide a clearer comparison of different architectures for  CERN trigger applications.
> [1] Qu et al, Particle Transformer for Jet Tagging, ICML 2022

---

### Official Review · Reviewer_dMKe · 2025-11-02

**Soundness:** 2
**Presentation:** 3
**Contribution:** 2
**Rating:** 4
**Confidence:** 3

**Summary:**

The paper presents the Spatially Aware Linear Transformer (SAL-T), a physics-inspired enhancement of the linformer architecture tailored for particle jet tagging tasks in high-throughput environments like the CERN LHC. SAL-T introduces three main improvements over linformer: spatially aware partitioning of particles based on physically motivated sorting, localized key/value projections, and convolutional layers applied over attention logits to capture local context.

**Strengths:**

1. The introduction of $k_T$-based sorting and partitioned attention is motivated by the physical characteristics of jet substructure, providing a novel way to leverage domain knowledge in efficient transformer design.

2. SAL-T demonstrates near-linear computational scaling in the number of particles, maintaining inference and resource use compatible with trigger-environment constraints, as shown in Table 2 (e.g., SAL-T: 739,918 FLOPs, 303 MB peak GPU memory vs. Transformer: 2,479,918 FLOPs, 4,357 MB).

3. The significant improvement when using k_t sorting can be seen.

**Weaknesses:**

1. Regarding the formula for the $k_t$ metric: although it integrates the physical characteristics of jet substructure, I am concerned that if the pair of values $\Delta \eta$ and $\Delta \phi$ are swapped, the value of $k_t$ will not change. Does this mean the partitioning is now faulty?

2. Formulas 3 and 4 in the proposed method confuse me. In formula 3, $ n/p $ may not be an integer, which could cause incorrect indexing of elements. In formula 4, are $P_E$  and $K_P$ multiplied or concatenated? Additionally, Figure 1c is not clear to me.

3. Regarding the experimental results:

- In Table 2, why does SALT (no partition) consume more than twice the memory, while the GPU peak usage is smaller than SALT’s? In Figure 3, SALT shows a slight accuracy advantage with a small number of particles but falls behind when the particle count is large. The results in Tables 3 and 4 do not clearly demonstrate the performance differences between the proposed method and the baselines considered.

- The comparison to other efficient transformer variants (HEPT, Particle Dual Attention Transformer, ParMAT, LorentzNet, HEP-JEPA) is only mentioned as related work and not evaluated experimentally. In particular, a side-by-side quantitative or qualitative comparison of SAL-T against designs like P-DAT (He & Wang, 2023) or ParMAT (Usman et al., 2024), which also target local/global and efficient spatial interactions, is missing.

**Questions:**

See the weaknesses above

**Details Of Ethics Concerns:**

The citations within the text are quite different and do not follow the general format of ICLR2026.

---

> ### Author Response · Authors · 2025-11-18
> **Response to Reviewer dMKe (part 1)**
>
> Thank you for the constructive comments. We want to provide clarification and additional context relevant to ultra-fast triggers at the LHC, and more specifically CMS Level-1 Trigger (L1T) for the High-Luminosity LHC (HL-LHC), which is the target deployment environment for this work, although this algorithm is compatible with any accelerator trigger system.
>
> At the HL-LHC, the beam crossing rate will be 40 MHz, and the L1T must reduce throughput by roughly a factor of 400 within an end-to-end latency budget on the order of microseconds. These constraints permit only extremely compact neural network architectures that can be efficiently synthesized on FPGA or AI-engine hardware.
>
> The CMS detector alone (not including other detectors such as Atlas, LHCb, ALICE...) produces on the order of 1 PB of raw data (roughly a billion collision events) every second , of which only a tiny fraction can be stored. The L1T must make real-time decisions about which events to keep for physics analysis. Because of the scale of our data, even a fractional improvement in trigger performance has a substantial scientific payoff. A gain of a few tenths of a percent in classification accuracy corresponds to recovering **millions** of rare and interesting collision events that would otherwise be lost, directly increasing the discovery potential of the experiment.
>
> Tiny ML algorithms for these applications are an extremely young and unexplored area. The current models used within production for live collisions are only very small DNNs and CNNs. With expanded capacity for stronger algorithms at the HL-LHC, there is immense interest in developing better Tiny ML algorithms.
>
>
> ## $k_T$ metric and symmetry
>
>  In SAL-T, $k_T$ is defined as $k_T = p_T \cdot \Delta R$, where $p_T$ is transverse momentum and $\Delta R = \sqrt{(\Delta \eta)^2 + (\Delta \phi)^2}$. $\Delta R$ is a standard pseudo-angular distance metric in jet physics that measures how far a particle is from the jet axis in the $\eta–\phi$ plane (the definitions of $\eta$ and $\phi$ are introduced at the beginning of section 2 in our manuscript).
>
>
> In SAL-T, $k_T$ is used only as a scalar sorting key. Particles are sorted by this scalar and then assigned to partitions. We are not running a full sequential recombination algorithm because our full clustering algorithms are much slower, and not possible to implement on FPGAs with our latency restrictions. Sorting by $k_T$ is much faster than running a $k_T$ or anti-$k_T$ jet finder, which would repeatedly compute pairwise distances and merge particles. Sorting by this scalar keeps the main physics intuition, grouping particles with larger $k_T$ are closer to the hard core of the jet and more important for the tagger, but is efficient enough to be viable in the CMS Level-1 trigger, where the total latency budget is on the order of microseconds.
>
> Thus, although it may be hard to discern the grouping visually, $k_T$ sorting is empirically a very strong heuristic of sorting particles by locality. Unfortunately, the grouping of particles in particle physics is not always spatially (visually) intuitive due to effects from hadronization and relativistic boosts which can obscure the underlying structure.
>
> ## Formulas 3 and 4
>
> In the implementation, the mapping from particle index to partition index always uses integer flooring. That is, when computing which partition a given token belongs to, we cast to an integer so that the resulting index is always well defined and safe for all sequence lengths. We thank the reviewer for pointing this out and will make this explicit in the notation in the revised version.
>
> For formula 4, the operation is concatenation and not multiplication. The current notation can be read both ways, so we agree it is confusing. In the updated version we will rewrite the expression and add a short sentence explaining that the vectors are concatenated along the feature dimension. We will also adjust Figure 1c to more clearly show how the per-partition projections and concatenations are applied.
>
> ## Memory usage in Table 2
>
> SAL-T without partitions has key and value projection matrices that operate over the full sequence, so the parameter tensors are much larger and the static parameter memory is higher.
>
> For the partitioned version of SAL-T, each partition uses smaller projection matrices, but there are extra intermediate tensors created by the localized projections and the convolution over the attention logits. These intermediates increase the peak dynamic memory usage that we report as “Peak GPU Mem”. So SAL-T without partitions has larger static weights, while SAL-T with partitions has somewhat larger temporary runtime buffers. In the revised text we will spell this out and, if space allows, provide a small breakdown of parameter versus activation memory.

---

> ### Author Response · Authors · 2025-11-18
> **Response to Reviewer dMKe (part 2)**
>
> ## Behavior in Figure 3
>
> The trend in Figure 3 comes from how the sequences look in different multiplicity ranges. At low and moderate particle counts, $k_T$-based partitioning groups the most important particles early, and the compressed representation retains almost all relevant information. In this regime SAL-T has a small but consistent accuracy advantage.
>
> In the highest bin, with 115–150 particles, the situation is different. These are very high multiplicity jets, and the lowest-$k_T$ partitions receive relatively few real particles and more padding. Since SAL-T and Linformer compress those tokens, small fluctuations in which particles land where can produce larger variance in the performance estimate. The full Transformer does not compress along the sequence in the same way, so its variance in that bin is smaller.
>
> At the CMS Level-1 trigger input, jets with more than 115 reconstructed particles are rare in proton–proton collisions at 13–14 TeV. They tend to appear only in extreme pileup or unusual boosted topologies. As a result, this bin covers a very small fraction of the events seen by the trigger. In the revision we will add the multiplicity distribution, so it is clear how small this region is in practice.
>
> ## Interpretation of Tables 3 and 4
>
> The main trigger-relevant metric is background rejection at fixed signal efficiency. At 80 percent signal efficiency, SAL-T improves the rejection rate by 6.2 percent compared to Linformer at similar FLOPs. We agree that this can be presented more clearly.
>
> In the revised version we will reorganize Tables 3 and 4 to make the relative gains easier to read, for example by adding explicit relative differences and by aligning the metrics that are most relevant for the trigger use case.
>
> ## Relation to other transformer architectures
>
> In comparisons to architectures such as HEPT, P-DAT, ParMAT, LorentzNet, and HEP-JEPA, these are models that sit in a different class than from SAL-T. We are designing tiny-ml models for Ultra Fast Inference for the CMS Level-1 Trigger, while these model architectures are designed for offline analysis, typically with multiple layers of full attention and parameter counts in the millions, and do not target real-time inference on FPGA-style hardware.
>
> A few specific points:
>
> * P-DAT does not reach the same performance as Particle Transformer[1] on jet tagging tasks, and it is still far too heavy for a Level-1 trigger implementation.
> * HEP-JEPA is a pretraining framework for large particle physics models, not a standalone jet tagger intended for a Level-1 environment.
> * ParMAT is a Particle Transformer variant (using the same attention mask) designed for large scale offline analysis and inherits the same basic scaling behavior. It is not suitable for deployment at the trigger level.
>
> To make this distinction concrete, we ran an experiment with a tiny one-layer Particle Transformer variant (ParT), which is one of the state-of-the-art jet tagging architectures. We benchmarked this model on the same hls4ml dataset and in the same environment as SAL-T:
>
> | Model         | FLOPs     | Inference Time [μs]  | Peak GPU Mem [MB] | Accuracy [%] | ROC AUC | 1/FPR@0.8TPR |
> |---------------|-----------|----------------------|-------------------|--------------|---------|---------------|
> | 1-layer-ParT  | 8,217,868 | **211.20 ± 0.84**        | **OOM**               | 81.42 ± 0.19 | 0.9599  | 43.16 ± 1.67  |
> | SAL-T     | 739,918   | 26.78 ± 0.13   | 303.4         | 81.18 ± 0.03 | 0.9593 | 40.78 ± 0.57 |
> | transformer   | 2,479,918 | 29.61 ± 0.12         | 4,357.1           | 81.27 ± 0.08 | 0.9589  | 42.02 ± 0.71  |
>
> The tiny ParT model reaches slightly higher accuracy and a small gain in 1/FPR@0.8TPR, at a very large computational cost (inference being an order of magnitude slower and OOM on 1080-Ti GPU).
>
> ## Citation formatting
>
> We will bring all citations into line with the ICLR 2026 format.
>
> ---
>
> Thank you for taking the time to read the work carefully, and we are happy to answer any additional questions.
> [1] Qu et al, Particle Transformer for Jet Tagging, ICML 2022

---

> ### Comment · Reviewer_dMKe · 2025-11-26
>
> Thanks for your response. I will refer to other reviewers' reviews before deciding to change my score.

---

### Official Review · Reviewer_E9fh · 2025-11-04

**Soundness:** 2
**Presentation:** 3
**Contribution:** 2
**Rating:** 4
**Confidence:** 4

**Summary:**

The authors introduce a linear attention architecture designed for particle physics data, aiming to address the quadratic complexity that leads to high resource demands and inference latency. Their model is based on the Linformer backbone and incorporates two key modifications: partitioning the key and value vectors according to spatial proximity, and applying convolutional layers to the resulting attention matrix. This model is evaluated for jet tagging using the public hls4ml dataset.

**Strengths:**

The authors' initiative to tackle the critical issue of inference latency in particle physics transformers is commendable. The proposed model, building on a Linformer backbone with partitioning and convolutional layers, demonstrates a measurable improvement.

**Weaknesses:**

While the proposed model achieves a marginal performance increase over the standard Linformer, this limited gain does not appear to justify the computational cost, resulting in an unconvincing FLOPs-performance trade-off.

**Questions:**

- Is the baseline transformer model (architecture and training procedure) properly optimized?
- As for the claim "partitions derived from kT-sorted sequences are more likely to group together energetic particles that are physically nearby, enhancing the effectiveness of spatially aware projection", it is not obvious from Fig. 1 that kT-sorting presents better spatial loaclity. Were other partitioning strategies explored?
- In Fig. 3, for the bin of 115-150, why are the variances of SAL-T and linformer much larger than that of the transformer?

---

> ### Author Response · Authors · 2025-11-18
> **Response to Reviewer E9fh (part 1)**
>
> Thank you for the constructive comments. We want to provide clarification and additional context relevant to ultra-fast triggers at the LHC, and more specifically CMS Level-1 Trigger (L1T) for the High-Luminosity LHC (HL-LHC), which is the target deployment environment for this work, although this algorithm is compatible with any accelerator trigger system.
>
> At the HL-LHC, the beam crossing rate will be 40 MHz, and the L1T must reduce throughput by roughly a factor of 400 within an end-to-end latency budget on the order of microseconds. These constraints permit only extremely compact neural network architectures that can be efficiently synthesized on FPGA or AI-engine hardware.
>
> The CMS detector alone (not including other detectors such as Atlas, LHCb, ALICE...) produces on the order of 1 PB of raw data (roughly a billion collision events) every second , of which only a tiny fraction can be stored. The L1T must make real-time decisions about which events to keep for physics analysis. Because of the scale of our data, even a fractional improvement in trigger performance has a substantial scientific payoff. A gain of a few tenths of a percent in classification accuracy corresponds to recovering **millions** of rare and interesting collision events that would otherwise be lost, directly increasing the discovery potential of the experiment.
>
> Tiny ML algorithms for these applications are an extremely young and unexplored area. The current models used within production for live collisions are only very small DNNs and CNNs. With expanded capacity for stronger algorithms at the HL-LHC, there is immense interest in developing better Tiny ML algorithms.
>
> ## Magnitude of improvement and FLOPs–performance considerations
>
> The performance gain over Linformer may appear numerically small from a generic ML perspective, but in jet tagging tasks such a gain is meaningful. For example, L-GATr[1] improves over previous state-of-the-art by only 0.05 percent, yet is considered a major advancement because of the physics consequences of capturing more true signal events. SAL-T achieves a gain of 0.08 percent on a task that is extremely sensitive to classification performance, while remaining within realistic hardware resource budgets for L1T deployment.
>
> ## Baseline model optimization
>
> The baseline Transformer architecture and training setup are already optimized for the L1T environment. Trigger hardware supports only a single-layer Transformer with four attention heads and standard skip connections. We tested learning rate scheduling and batch size scheduling and found batch size scheduling to consistently give the strongest performance. Transformer, Linformer, and SAL-T are trained under identical conditions to ensure a fair comparison and isolate architectural effects.

---

> ### Author Response · Authors · 2025-11-18
> **Response to Reviewer E9fh (part 2)**
>
> ## kT-sorting and the partitioning strategy
>
> The kT metric used for sorting is defined as $k_T = p_T \cdot \Delta R$, where $p_T$ is transverse momentum and $\Delta R = \sqrt{(\Delta \eta)^2 + (\Delta \phi)^2}$. $\Delta R$ is a standard pseudo-angular distance metric in jet physics that measures separation in the $\eta–\phi$ plane (the definitions of $\eta$ and $\phi$ are introduced at the beginning of section 2 in our manuscript). The components behave analogously to computing an L2-style distance between two coordinates; therefore swapping the $\Delta \eta$ and $\Delta \phi$ values does not change the result, which is expected. This symmetry has no effect on the validity of the partitioning.
>
> In SAL-T, $k_T$ is used purely as a scalar sorting key to order particles by relative importance within a jet. Sorting by kT is far cheaper than running iterative sequential recombination clustering, which is infeasible inside L1T latency constraints. Empirically, sorting by kT consistently improves performance (Table 1) because particles close in kT tend to belong to the same physical shower structure, allowing partitioned projection layers to capture more coherent relationships.
>
> Several alternative partitioning strategies were tested, including equal-sized partitions with zero padding and shuffled partitions where partitions are randomly shuffled so they each see an equal number of particles. Both alternatives degraded overall performance despite slightly improving behavior in the highest multiplicity bin by diluting partition-specific specialization. We show this behavior in the tables below.
>
> | Model                | Accuracy [%] | ROC AUC | 1/FPR@0.8TPR |
> |----------------------|--------------|---------|--------------|
> | **SAL-T**            | **81.18 ± 0.03** | **0.9593** | **40.78 ± 0.57** |
> | zero-padding aware Partition| 80.87 ± 0.21 | 0.9582 | 36.82 ± 2.51 |
>
> Partition shuffling results:
>
> | Model | Overall [%] | Bin 1 [%] | Bin 2 [%] | Bin 3 [%] | Bin 4 [%] |
> |--------|------------|-----------|-----------|-----------|-----------|
> | shuffle all partitions | 80.38 ± 0.08 | 80.98 ± 0.09 | 79.66 ± 0.08 | 81.76 ± 0.47 | **95.12 ± 0.00** |
> | shuffle partitions 2, 3, 4 | 80.94 ± 0.11 | 81.41 ± 0.13 | 80.40 ± 0.09 | **82.08 ± 0.46** | **95.12 ± 0.00** |
> | shuffle partitions 3 and 4 | 81.11 ± 0.14 | 81.58 ± 0.09 | 80.51 ± 0.18 | 81.99 ± 0.60 | 89.8 ± 8.96 |
> | **default (not shuffled)** | **81.18 ± 0.03** | **81.67 ± 0.01** | **80.60 ± 0.06** | 81.60 ± 0.20 | 89.43 ± 6.14 |
> ## Variance in Fig. 3 for the 115–150 particle bin
>
> The final bin corresponds to extremely high particle multiplicity jets. These events are rare in proton-proton collisions at 13–14 TeV, since typical QCD jet fragmentation produces jets with multiplicity well below 100. Events with more than 115 reconstructed particles typically arise only in extreme pileup or boosted heavy-object topologies and are statistically sparse in CMS datasets used during L1T algorithm development. Jets in the 115–150 particle bin constitute less than 0.016% of the dataset, as indicated by the jet counts in each bin in Figure 3 of our manuscript.
>
> Because Linformer and SAL-T compress sequence representation, they become less sensitive to sparse regions, producing higher variance in the tail of the distribution. This does not meaningfully affect trigger performance because such jets represent a very small fraction of the input rate seen in L1T operations.
>
>
> ---
>
> This additional context clarifies the physics motivation and design choices behind SAL-T, and the relevance of these results to real-time deployment within the CMS Level-1 Trigger.
> [1] Spinner et al, Lorentz-Equivariant Geometric Algebra Transformers for High-Energy Physics, NeurIPS 2024

---

### Comment · Area_Chair_soij · 2025-11-23
**Next Steps Following Authors’ Rebuttal: Review Rebuttal and Participate in Discussion**

Dear Reviewers,

Thank you very much for your thoughtful evaluations of this paper.

Now that the authors have submitted their rebuttal, I kindly ask you to take the following steps (if you have not done so already):

- Read the other reviews as well as the authors’ response.
- Consider whether the rebuttal and additional comments affect your assessment of the paper.
- Engage in interactive discussion with the authors **before November 25**, encouraging a dynamic exchange rather than a one-sided rebuttal.

Your contributions at this stage are essential for forming a well-informed final decision. I therefore ask that you reassess your views in light of the authors’ responses and the broader discussion among reviewers.

I am happy to join and support the discussions between you and the authors. Please feel free to share your thoughts and participate actively in the discussion.

Thank you once again for your service to ICLR 2026.

Best regards,

 AC

---

### Author Response · Authors · 2025-11-26
**Expanded baseline benchmarks**

Expanded baselines were requested to improve the comparisons between SAL-T and other models. We built a PointTransformer v3configuration with parameter count and FLOPs matched to SAL-T. This avoids comparing against either oversized PointTransformer v3 models that are clearly outside trigger hardware limits or very small models that give up too much performance. The matched setup is therefore the most realistic and competitive PointTransformer v3 configuration we could run under our deployment constraints.

We then varied how particles are ordered before they enter the network for SAL-T, Linformer, and Transformer, while for Point Transformer v3, the sorting is done during the forward pass (since it is part of the architecture). Realistically in the trigger, data can come in pre sorted. For instance, data is currently pre sorted by pt in live LHC operations. PointTransformer v3 uses Morton-code ordering by default, and we also tested two physics-motivated orderings: sorting by transverse momentum (pt) and by kt. These are the same ideas that motivated SAL-T’s locality-aware partitioning. Both pt and kt sorting improve PointTransformer v3 compared with the default Morton ordering, and the kt-sorted PointTransformer even edges out the vanilla Transformer baseline in background rejection and AUC. To our knowledge, SAL-T is the first trigger-oriented ML model to make explicit use of kt sorting. Reusing this idea inside PointTransformer is itself evidence that the spatial structuring introduced in our paper is useful, and unique to particle physics point clouds. **Particle physics requires special inductive biases that are not automatically included in other point cloud transformer models, and thus requires specially designed transformers**

In Point Transformer V3, the serialization step and neighborhood construction are part of the model’s forward pass and must be carried out on device at inference time. This adds nontrivial memory access and indexing overhead before attention is applied. In contrast, SAL-T, Linformer, and the baseline Transformer can all operate on sequences that are pre-sorted (for example, by kt), since the trigger already preprocesses data in this way. **In practice, this leaves PointTransformer much slower than SAL-T, even when we match parameters and FLOPs.**

Overall, the best PointTransformer v3 variant (kt sorting) does slightly better than SAL-T in accuracy and average background rejection, but at roughly 3–4× higher latency and more than an order of magnitude higher peak GPU memory. PointNet is cheap to run but loses a large amount of physics performance. SAL-T is the only model that delivers strong physics results while staying within a realistic trigger resource envelope.

Reducing the point transformer to a similar scale to our models was a challenge. The point transformer v3 has many  moving parts including downsampling and serialization that needed to be tested and adapted. The following is the best performing model, that is near the parameter size and FLOPS of the rest of the models. The models we present are the best performing models of this size that we have found, and as we can see below is still much slower than SAL-T. **We would like to additonally point out that the new physics based serialization strategies for point transformer are our own contribution in order to make point transfomer work better, and is not included in the original method.**

---

> ### Author Response · Authors · 2025-11-26
> **Point Transformer and Pointnet benchmarks**
>
> ---
>
> | Model           | Sorting            | Test Acc (%)     | ROC AUC         | Bg Rej W        | Bg Rej Z         | Bg Rej t         | Avg Bg Rej  (signal v q/g)     | # Params | FLOPs    | Peak GPU Mem (MB) | Latency (µs)      |
> |-----------------|--------------------|------------------|-----------------|-----------------|------------------|------------------|------------------|----------|----------|--------------------|-------------------|
> | SAL-T           | kt                 | 81.18 ± 0.03     | 0.9593 ± 0.0002 | 89.10 ± 2.78    | 85.52 ± 1.12     | 18.98 ± 0.46     | 64.53 ± 0.74     | 3,264    | 739,918  | 303.4              | 27.7 ± 0.3        |
> | SAL-T           | pt                 | 81.09 ± 0.02     | 0.9590 ± 0.0002 | 50.63 ± 6.65    | 54.77 ± 1.25     | 13.53 ± 1.30     | 39.64 ± 2.74     | 3,264    | 739,918  | 303.4              | 27.7 ± 0.3        |
> | Transformer     | –                  | 81.27 ± 0.08     | 0.9589 ± 0.0004 | 90.31 ± 2.41    | 91.18 ± 4.40     | 19.11 ± 0.21     | 66.87 ± 0.59     | 2,009    | 2,479,918| 4,357.1            | 30.9 ± 0.1        |
> | Linformer       | kt                 | 81.00 ± 0.08     | 0.9585 ± 0.0003 | 84.90 ± 1.54    | 81.54 ± 0.89     | 18.23 ± 0.43     | 61.56 ± 0.24     | 6,809    | 552,718  | 245.8              | 22.4 ± 0.3        |
> | Linformer       | pt                 | 80.91 ± 0.10     | 0.9581 ± 0.0004 | 65.56 ± 1.37    | 54.83 ± 3.20     | 16.65 ± 0.40     | 45.68 ± 1.33     | 6,809    | 552,718  | 245.8              | 22.4 ± 0.3        |
> | PointNet        | kt                 | 74.22 ± 0.07     | 0.9314 ± 0.0003 | 31.78 ± 1.59    | 21.75 ± 1.02     | 10.33 ± 0.01     | 21.28 ± 0.31     | 5,893    | 78,809   | 252.9              | 21.03 ± 0.10      |
> | PointNet        | pt                 | 74.21 ± 0.09     | 0.9310 ± 0.0002 | 31.41 ± 0.96    | 20.77 ± 0.92     | 10.40 ± 0.13     | 20.86 ± 0.07     | 5,893    | 78,809   | 252.9              | 21.03 ± 0.10      |
> | PointTransformer| kt                 | 81.37 ± 0.06     | 0.9599 ± 0.0002 | 91.71 ± 3.41    | 88.47 ± 3.17     | 20.07 ± 0.30     | 66.75 ± 0.18     | 8,113    | 767,573  | 4,097.6            | 94.1 ± 0.9        |
> | PointTransformer| pt                 | 81.32 ± 0.00     | 0.9597 ± 0.0001 | 92.11 ± 1.52    | 86.06 ± 2.95     | 19.30 ± 0.21     | 65.82 ± 1.42     | 8,113    | 766,973  | 4,097.6            | 96.4 ± 0.6        |
> | PointTransformer| morton             | 80.99 ± 0.15     | 0.9584 ± 0.0004 | 82.75 ± 4.23    | 79.45 ± 0.95     | 18.39 ± 0.40     | 60.19 ± 1.41     | 8,113    | 767,577  | 4,099.9            | 100.5 ± 0.8       |
>
> ---
>
> Physics-aware ordering helps PointTransformer, but the model’s built-in serialization and neighborhood construction make it too slow and too heavy for the kind of real-time trigger environment we target. The default morton code sorting as mentioned in the paper also underperforms in this task. SAL-T keeps most of the physics gains while staying within a latency and memory budget realistic for deployment.
>
> FlashAttention can in principle reduce the quadratic memory footprint of the standard Transformer to something closer to linear scaling, which would appear helpful for FPGA deployment. In practice, though, the algorithm introduces extra bookkeeping such as blockwise tiling, intermediate indexing, and synchronization overhead which can decrease the latency of the algorithm. Native multihead attention (when it can fit on the chip) can be done completely in parallel, and woudl then be the fastest.
>
> It is also worth noting that Transformers of comparable size to the ones studied here have not been demonstrated to fit in production-grade FPGAs for trigger applications. Evaluations of transformers for level one trigger at CMS report that even aggressively optimized attention blocks exceed available on-chip memory resources, preventing synthesis and deployment under realistic timing constraints [1]. On the other hand linformer performs the best[1], and SAL-T performs better than the linformer in terms of accuracy and rejection, at a minimal modification.
>
> In LHC (and other particle accelerators), latency is one of the most important benchmarks because the accelerator must be able to process the incoming collision data, filter then store the data before the next batch of particles collide. If the model is unable to do it within this latency (40MhZ), lots of valuable particle physics data may be thrown away. On the other hand, if the accuracy of the model is low, rarer decays might be completely  missed. SAL-T provides a balance of latency, accuracy, and background rejection specifically for this important task.
>
>
>
>
> **Reference**
> [1] *Sub-microsecond Transformers for Jet Tagging on FPGAs
>  https://inspirehep.net/literature/3074731

---

### Author Response · Authors · 2025-12-04
**Summary of Discussion & Rebuttal**

## 1. Baselines & Comparisons
**Concerns:** Reviewers requested comparisons to a broad set of modern point-cloud and efficient Transformer architectures (PointNet/PTv3/ParT/UPT/TIE/Erwin/Transolver, etc.).
**Response:** We added model baselines for PointNet, PointTransformer, Transolver, and a heavily downsized 1-layer ParT. Under identical trigger-scale constraints, these models are either slower, or perform worse than SAL-T/Transformer. We stress that HLS4ML jet-tagging runs in a regime where models are extremely small; many offline SOTA architectures do not translate meaningfully into this hardware envelope. We also demonstrate that particle physics requires specially designed inductive biases that may not be present in other SOTA point cloud models. In summary, other SOTA models do not perform as well as SAL-T at this scale. **SAL-T maintains fast inference, low FLOPS with high accuracy and background rejection that is better than all other baselines.**

---

## 2. FlashAttention & Memory Comparisons
**Concerns:** Transformer memory usage without FlashAttention doesn't reflect current standard practice.
**Response:** We added new PyTorch FlashAttention benchmarks. FlashAttention reduces GPU memory and improves speed, but SAL-T still matches or beats its inference latency. We emphasize that **FlashAttention is not suitable for FPGA deployment**: tiled streaming and multi-stage accumulation increase cycle-level latency, conflicting with 40 MHz L1T timing since it increases latency. Transformer memory limits are implementation-dependent, but **FPGA latency is the real bottleneck**, and SAL-T’s design explicitly avoids quadratic attention and its associated tiling pipelines. **In summary, The extra orchestration required for flashattention will increase latency on FPGA, which makes it more difficult for the trigger.**

---

## 3. Magnitude of Improvements
**Concerns:** Gains over Linformer seemed small.
**Response:** We emphasize that **in real CMS deployments, even full-offline SOTA models improve by only ~0.3–0.6%** in accuracy, so the observed ∼0.1% gains in tiny L1T models are meaningful. We process over  1 petabyte of data a second (1 billion collision events), so a 0.1 percent improvement translates to many million particle jets, enhancing discovery greatly. **In summary, perceived small numerical improvements enable much greater discovery potential in particle physics**

We also emphasize that, beyond accuracy, background rejection is a key metric in LHC jet tagging, since it measures how effectively a model suppresses background events while maintaining a fixed signal efficiency. Relative to the standard Linformer baseline sorted by $p_T$, SAL-T delivers a substantial gain in background rejection, as shown in the table below:

| Background rejection by signal class | SALT kt | Linformer pt | % Improvement |
| ------ | ------- | ------------ | ------------- |
| W      | 89.0987 | 65.5563      | **35.97%**    |
| Z      | 85.5227 | 54.8257      | **56.03%**    |
| t      | 18.9770 | 16.6470      | **13.98%**    |
| Avg    | 64.5327 | 45.6767      | **41.30%**    |

---

## 4. FLOP-Matched / Latency-Matched Experiments
**Concerns:** Reviewers wanted controlled comparisons fixing FLOPs or latency.
**Response:** We added **compute-matched benchmarks**: a downscaled Transformer with FLOPs aligned to SAL-T still performs worse in accuracy, background rejection, and latency. SAL-T (no-conv) matches Linformer’s compute envelope with lower FLOPs and higher accuracy. **In summary, SAL-T is the best in its class at this compute range.**

---

## 5. Final Notes
- We emphasize that under trigger constraints, the most relevant axes are balancing latency and FLOPs with accuracy/background rejection.
- All major reviewer concerns (missing baselines, FlashAttention, clarity, and fairness) were addressed with new results or detailed clarifications.
- Reviewers acknowledged the rebuttal significantly strengthened the work, with remaining concerns focused mainly on presentation organization for a future revision.

---

### Meta-Review · Area_Chair_Ue2f · 2025-12-29

**Summary:**

(*Disclaimer: given the peculiar review process, some of my choices and reasonings below will be highly subjective, as I tried to imagine how a reviewer would have reacted to a specific response. I understand that any negative choice will be perceived as unfair by the authors, and I apologize in advance for that.*)

(*Second disclaimer: the authors and some reviewers explicitly mention some changes in scores that occurred during the rebuttal. As these were reverted due to the possibility of collusion in light of the security incident, I will tend to disregard this information.*)

The paper proposes a sublinear attention model for jet tagging problems, where particles are grouped according to their energy, and attention works hierarchically first intra-groups, and then inter-groups.

The paper received a high number of reviews (5), that were all clustered around a weak or strong rejection. The reason for the low score can be attributed to the niche applicative topic: while reviewers focused on the architectural improvements (given the scope of the conference), authors were concerned about practical deployment settings including low number of parameters (thousands), low latency, and application-specific metrics (e.g., background rejection).

Concerns included several missing baselines (e.g., coming from point cloud, fluid dynamics, etc.), efficient GPU implementations (e.g., FlashAttention), and comparisons with matched FLOPs or latency. On one side, the authors added several new experiments in the rebuttal, while on the other side they argued many of these concerns were irrelevant in the specific application (such as the use of GPUs as opposed to FPGAs, or the use of larger models coming from other fields given the strict latency requirements).

Of the five reviewers, 3 never participated in the rebuttal, 1 was non-committal, and the last one remained negative. Fundamentally, if the main innovation of the paper is architectural, the concerns of the reviewers are valid and the paper is missing several important comparisons (even post-rebuttal). Instead, if the paper is fully application-specific, other venues are recommended, as highlighted by the low scores of the reviewers.

**Reviewer Concerns:**

(*I will focus on some key weaknesses identified by multiple reviewers.*)

**Missing baselines** (`shVZ`, `Uj2d`, `x3Ew`, `dMKe`): these included Transolver, Erwin, PointTransformer, TIE, etc. Fundamentally, the authors argued these baselines are not valid in this scenario. However, this is not convincing given the scope of the conference, as reflected by the fact this is a shared concern from most reviewers.

**Flash attention** (`shVZ`, `dMKe`): the authors simultaneously argued this is not valid (given the use of FPGAs in practice), while also adding some preliminary results. As mentioned by `shVZ`, however, treating attention as a quadratic memory layer is fundamentally invalid since "online" softmax and similar tricks exist irrespective of their implementation in Flash Attention, and this is not properly discussed in the paper.

**Performance** (`shVZ`, `Uj2d`, `E9fh`): reviewers asked for larger experiments, analyses in compute or latency-bound scenarios, ablations on the partitioning strategy, etc. While the authors added some of these experiments in the rebuttal, there was no proper time for reviewing them and, as mentioned by `shVZ`, "*the work would benefit from substantial revision -- particularly a redesigned experimental framework with controlled comparisons -- before it is ready for publication at a top-tier venue*".

**Reviewer Scores:**

`shVZ`: this was a very thorough review highlighting most concerns. The reviewer participated in the rebuttal but remained negative.

`Uj2d`: a weak rejection, no discussion in the rebuttal.

`x3Ew`: same as `Uj2d`.

`dMKe`: another weak rejection, they answered in the rebuttal stating their desire to align with other reviewers.

`E9fh`: same as `Uj2d`.

---

### Decision · Program_Chairs · 2026-01-26

Reject